# Biomimetic chiral hydrogen-bonded organic-inorganic frameworks

Jun Guo [1] ✉, Yulong Duan[1], Yunling Jia[2], Zelong Zhao[1], Xiaoqing Gao[3], Pai Liu[2], Fangfang Li[1], Hongli Chen[2], Yutong Ye[2], Yujiao Liu [1], Meiting Zhao [4] ✉, Zhiyong Tang [5] ✉ & Yi Liu [1] ✉

Assembly ubiquitously occurs in nature and gives birth to numerous functional biomaterials and sophisticated organisms. In this work, chiral hydrogen-bonded organic-inorganic frameworks (HOIFs) are synthesized via biomimicking the self-assembly process from amino acids to proteins. Enjoying the homohelical configurations analogous to $\alpha$-helix, the HOIFs exhibit remarkable chiroptical activity including the chiral fluorescence ($g_{lum} = 1.7 \times 10^{-3}$) that is untouched among the previously reported hydrogen-bonded frameworks. Benefitting from the dynamic feature of hydrogen bonding, HOIFs enable enantio-discrimination of chiral aliphatic substrates with imperceivable steric discrepancy based on fluorescent change. Moreover, the disassembled HOIFs after recognition applications are capable of being facilely regenerated and self-purified via aprotic solvent-induced reassembly, leading to at least three consecutive cycles without losing the enantioselectivity. The underlying mechanism of chirality bias is decoded by the experimental isothermal titration calorimetry together with theoretic simulation.

As the natural gift, chiral matters and structures play an essential role in the life as well as the universe. Spanning over the length scale from subangstrom to light-year, spin electrons, amino acids, DNAs, proteins, snail shells, screw propellers, and nebulas all feature with chirality[1–3]. The exploitation of functional chiral materials has also long been a pivot of many fields covering medicine[4,5], catalysis[6,7], separation[8,9], biology[10,11], recognition[12,13], and optics[14–17]. Assembly is expected to be one of the most intriguing strategies to build the chiral hierarchies with premade arrangements, properties, and functionalities[18–22]. Therefore, the assembles of the ordered connection and harmonious communication between building blocks are expected to generate the collective property and synergistic functionality beyond those of individuals[23–25]. Nevertheless, one grand challenge for constructing chiral structures and materials lies in the high entropic barriers

associated with the chiral building blocks mostly of asymmetric geometries, nonplanar structures, and multiple conformations[26,27], which are hardly overcome by the enthalpic compensation of typical weak driving forces (e.g., hydrogen bonding, Van der Waals and π-π interactions)[28–30]. As a result, the collective performances of chiral assemblies are rarely reported along with chirality-related applications despite that there have been many examples such as chiral gels[31–33], polymers[34,35], and aggregators[36–38] of disordered structures.

Fortunately, nature teaches the lessons involving enantioselective evolution via self-assembly processes[39,40]. For instance, amino acids first connect into the right-handed $\alpha$-helices through enthalpy-favored peptide condensation[41], and then assemble into $\beta$-sheet via self-complementary hydrogen bonding (Fig. 1a)[42,43]. Distinct from discrete amino acid as well as single peptide chain, the assembled $\beta$-sheet is

[1]State Key Laboratory of Separation Membranes and Membrane Processes, School of Chemistry, Tiangong University, 300387 Tianjin, P. R. China. [2]School of Materials Science and Engineering, Tiangong University, 300387 Tianjin, P. R. China. [3]Wenzhou Key Laboratory of Biomaterials and Engineering, Wenzhou Institute, University of Chinese Academy of Sciences, 325000 Wenzhou, P. R. China. [4]Tianjin Key Laboratory of Molecular Optoelectronic Sciences, Department of Chemistry, Institute of Molecular Aggregation Science, Tianjin University, 300072 Tianjin, P. R. China. [5]CAS Key Laboratory of Nanosystem and Hierarchical Fabrication, CAS Center for Excellence in Nanoscience, National Center for Nanoscience and Technology, 100190 Beijing, P. R. China. ✉e-mail: junguo@tiangong.edu.cn; mtzhao@tju.edu.cn; zytang@nanoctr.cn; yiliuchem@whu.edu.cn

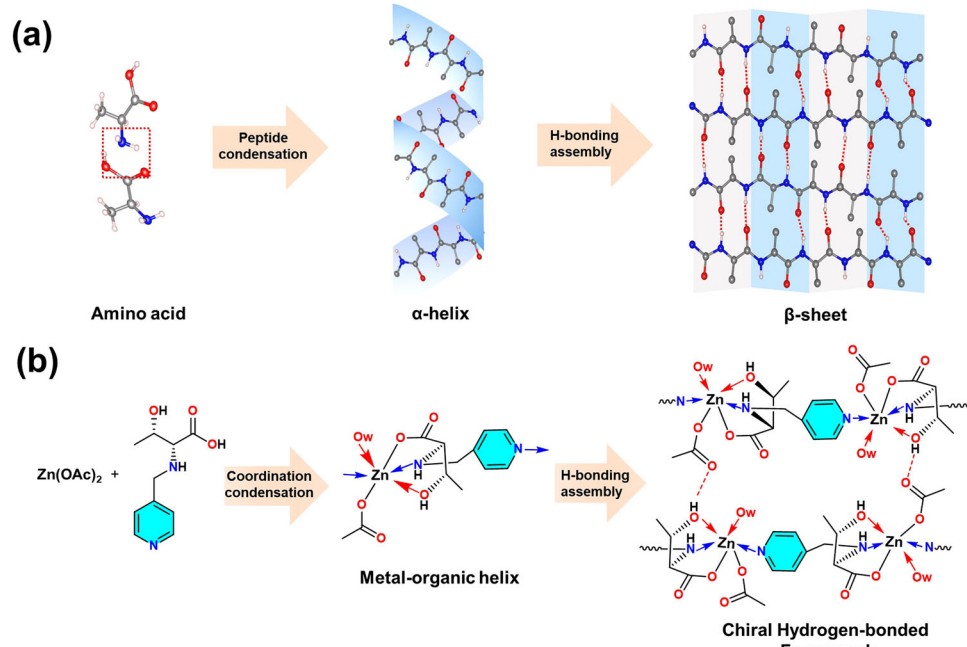

**Fig. 1 | Biomimetic hierarchical assembly strategy proposed in this work.**
**a** Schematic diagram of assembly evolution of natural proteins from chiral amino acids. The red dashed box represents the binding site involved in peptide condensation. **b** Structural diagram of chiral HOIFs proposed in this work via bioinspired assembly process. The blue arrow indicates the coordination of pyridyl units. The red dotted lines represent hydrogen bondings.

henceforth endowed with emerging hierarchical chirality and allosteric property[44,45].

Herein, inspired by this step assembly of proteins, we propose designing and fabricating a series of chiral hydrogen-bonded organic-inorganic frameworks (HOIFs) (Fig. 1b). In brief, chiral ligands and metal ions would combine into one-dimensional (1D) helical chains via enthalpy-favored coordination condensation. Sequentially, multiple complementary hydrogen bonds might facilitate the assembly of helical chains into two-dimensional (2D) grids or three-dimensional (3D) frameworks of well-defined chiral structures. Distinct from conventional chiral metal-organic frameworks (MOFs) and covalent organic frameworks (COFs), the reported chiral HOIFs have exhibited unique assembled-induced chiroptical activities including the emerging circular polarized luminescence (CPL). Also, chiral HOIFs could be reversibly disassembled into single helical strands and be further reassembled into frameworks without losing ordered arrangement. Hence, they have been utilized as recoverable chiral platforms for enantioselective recognitions, which would be not implemented by chiral MOFs and COFs formed via strong linkages.

## Results

### Preparation and characterization of chiral HOIFs

Enantiomeric *N*-(4-pyridylmethyl)-*d*(*l*)-threonine (abbreviated as *d*(*l*)-thr) derived from the biologic *d*(*l*)-threonine were first synthesized as chiral ligands and fully characterized (Supplementary Figs. 1–7 and Supplementary Table 1). Then, Zn(*d*(*l*)-thr)(CH$_3$COO)H$_2$O bulk crystals, denoted as [*d*(*l*)-Zn-HOIF BCs], were cultivated in an aqueous solution containing *d*(*l*)-thr and Zn(CH$_3$COO)$_2$·2H$_2$O at room temperature. Single-crystal X-ray diffraction (SXRD) analyses show that both *d*-Zn-HOIF BCs (CCDC NO. 2209408) and *l*-Zn-HOIF BCs (CCDC NO. 2209409) belong to the orthorhombic system and the chiral space group of P2$_1$2$_1$2$_1$, but are of opposite configurations (Supplementary Tables 2–13). Hence, we take *d*-Zn-HOIF BCs as the representative to describe the structure evolution (corresponding descriptions on *l*-enantiomer are available in Supplementary Fig. 8). In detail, the Zn(II)

center adopts a distorted octahedron with one pyridyl nitrogen (N1) from one *d*-thr, one amino nitrogen (N2), one carboxyl oxygen (O2) and one hydroxyl oxygen (O3) from another *d*-thr, and additional one acetate oxygen (O4) and one water oxygen (O6$_w$) (Fig. 2a). On account of the asymmetric coordination fashion of Zn(II) center, the associated [Zn(*d*-thr)(CH$_3$COO)H$_2$O]$_\infty$ chain is engendered via coordination condensation and presents a left-handed 2$_1$ screw with a pitch of 18.4 Å along the *c*-axis (Fig. 2b), analogously to the well-known α-helix structure arising from peptide condensation. Resembling the hydrogen-bonded assembly process of β-sheet from α-helix, 2D sheet-like structure within the *b*-*c* plane is spontaneously formed from above 1D chains via directional hydrogen bonding between the hydroxyl hydrogen with the uncoordinated acetate oxygen (O-H···O-C, 1.880 Å, Fig. 2c). Additionally, two pairs of complementary hydrogen bonding between coordinated water hydrogen and amine hydrogen with two acetate oxygens (O$_w$-H···O-C, 1.902 Å and N-H···O-C, 1.919 Å, Fig. 2d) from adjacent sheets facilitate eclipsed AA stacking of those chiral sheets into the final 3D framework along the *a*-axis. The powder X-ray diffraction (PXRD) of *d*(*l*)-Zn-HOIF BCs exhibits the same diffraction pattern as simulated one based on resolved single-crystal structure, with the first five peaks (i.e. 7.81°, 9.62°, 11.43°, 12.34° and 13.25°) assigned to the (011), (002), (012), (020) and (021) facet, respectively (Fig. 3a). Besides the crystallographic characterization, the nearly identical Fourier transform infrared (FT-IR) spectra (Supplementary Fig. 9) and thermogravimetric analysis (TGA) curves (Supplementary Figs. 10, 11) of *d*(*l*)-Zn-HOIF BCs also support their same coordination environment and equivalent thermal stability. The proposed biomimetic assembly process is also applicable to varied types of chiral HOIFs, as demonstrated by successful constructions of isoreticular *d*(*l*)-Ni-HOIFs and *d*(*l*)-Co-HOIFs following the identical diffraction patterns to *d*(*l*)-Zn-HOIFs (Supplementary Fig. 12). In order to obtain the exact cell parameters, PXRD refinements (Supplementary Fig. 13) were further done with satisfactory agreement. As results summarized in Supplementary Table 14, those isoreticular HOIFs display nearly the same crystallographic parameters but with slightly shrunken

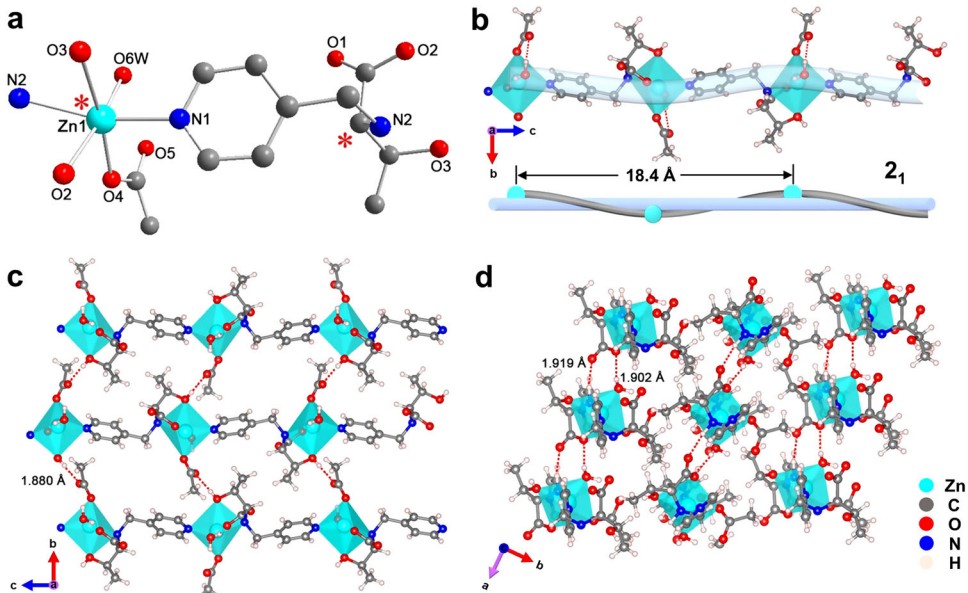

**Fig. 2 | Assembly evolution of *d*-Zn-HOIF characterized by SXRD. a** Asymmetric coordination mode of *d*-Zn-HOIF. The red asterisks represent chiral centers. **b** Assembled 1D helical chain with left-handed 2₁ screw and pitch of 18.4 Å along *c*-axis. **c** 2D grid. **d** 3D framework via complementary hydrogen bonding. The red dotted lines represent hydrogen bondings.

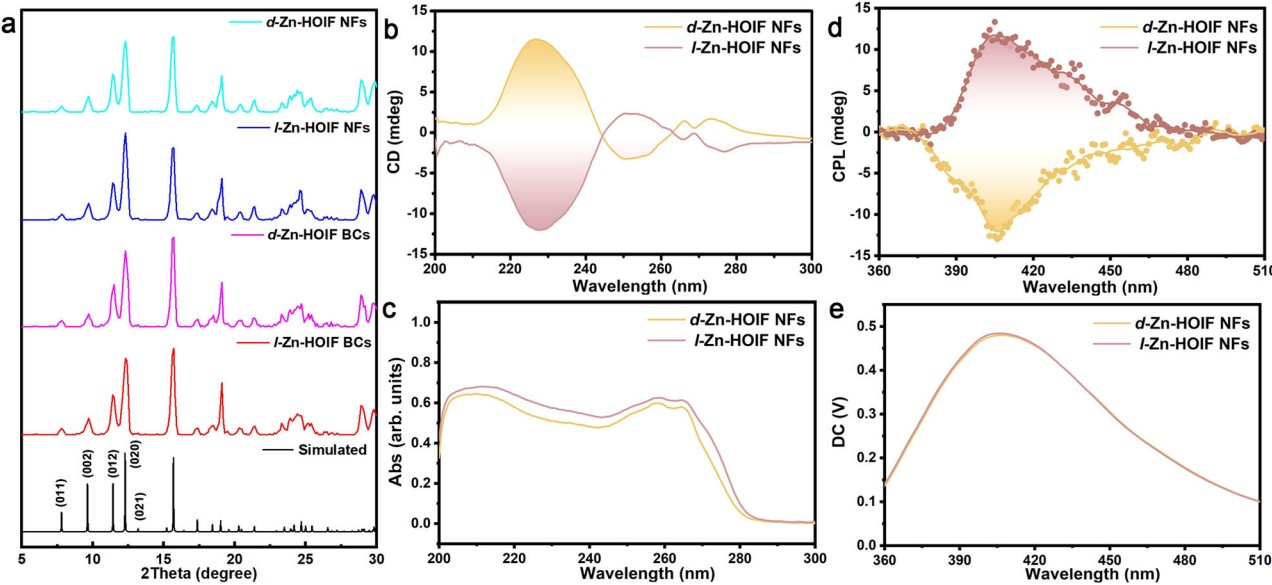

**Fig. 3 | Characterization of *d(l)*-Zn-HOIFs. a** PXRD patterns of *d(l)*-Zn-HOIF NFs and BCs along with the simulated one. **b** Solid CD and **c** UV absorption spectra of *d(l)*-Zn-HOIF NFs. **d** CPL spectra of *d(l)*-Zn-HOIF NFs excited at 265 nm. **e** DC value standing for normal fluorescence intensity of *d(l)*-Zn HOIF NFs.

cell dimensions for *d(l)*-Ni-HOIFs due to the smaller Ni²⁺ radius than others.

In order to take advantage of higher surface area and better solvent dispersibility, the corresponding *d(l)*-Zn-HOIF nanofibers (NFs) were fabricated by following our previously reported microemulsion confinement protocol[46]. The XRD patterns of *d(l)*-Zn-HOIF NFs were consistent with the diffraction pattern of *d(l)*-Zn-HOIF BCs and the simulated ones, demonstrating their successful preparations (Fig. 3a). Importantly, in comparison to hundreds of micrometers of *d(l)*-Zn-HOIF BCs (Supplementary Figs. 14, 15), the scanning electron microscopy (SEM) images show that the dimension of as-obtained *d(l)*-Zn-HOIF NFs greatly shortens to only ca. 200 nm in diameter (Supplementary Figs. 16, 17).

The chiroptical activity of *d(l)*-Zn-HOIF NFs was then explored by circular dichroism (CD) spectroscopy. The *d(l)*-Zn-HOIF NFs present strong and mirror-symmetric CD signals (Fig. 3b) but with nearly identical UV absorbance (Fig. 3c) at the wavelength from 200 to 300 nm, a typical indicator of the enantiomeric relationship. Specifically, the CD peak centered at 230 nm is assigned to the n → π* transition of the carboxyl chemophore[47]. While a serial of CD signals in the range of 260 to 280 nm are ascribed to the π→π* and n → π* transitions of rigid pyridyl ring[10,47] involved in the unique homo-helical configuration among *d(l)*-Zn-HOIF NFs. Noteworthily, the CD response acquired by *d(l)*-Zn-HOIF NFs is obviously distinct from the isolated *d(l)*-thr ligands as well as *d(l)*-thr-Zn complex (Supplementary Fig. 18). Moreover, though chiroptical activity in the visible

region is not observed for *d(l)*-Zn-HOIF NFs (Supplementary Fig. 19) due to the forbidden ligand-to-metal charge transfer (LMCT), isoreticular *d(l)*-Co-HOIF NFs and *d(l)*-Ni-HOIF NFs both display strong and mirror-symmetric CD response ranging from 200 to 800 nm (Supplementary Figs. 20, 21). Inspired by the observed intriguing chiroptical absorptions, we further wonder whether the chiral luminescence stemming from pyridyl units emerges, which has been

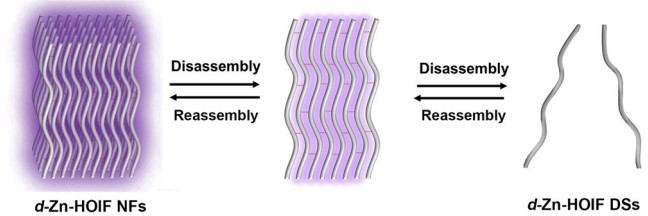

**Fig. 4 | Disassembly and reassembly conjecture of *d*-Zn-HOIF.** Schematic diagram of *d*-Zn-HOIF NFs with disassembly-reassembly dependent fluorescence. The red dotted lines represent directional hydrogen bondings.

not reported yet among the hydrogen-bonded frameworks. Impressively, the circularly polarized luminescence (CPL) spectrum of *d*-Zn-HOIF NFs (Fig. 3d) exhibits considerably negative CPL signals in the window of 360 to 510 nm. And, similarly strong but positive CPL signals are discerned in the same wavelengths for *l*-Zn-HOIF NFs of the opposite chiral configuration. In relation to absorption constrained within the UV region, the observed red-shifted CPL for *d(l)*-Zn-HOIF NFs is reasoned to the prominent charge transfer-induced dissipation[48,49] and thermal relaxation of excited states[50,51] (Supplementary Figs. 22–24). On basis of the identical normal luminescence (Fig. 3e), the anisotropy factor ($g_{lum}$) of CPL is calculated to be $1.7 \times 10^{-3}$ for both *d(l)*-Zn-HOIF NFs at the peak wavelength of 405 nm, a pretty high value even compared with the popular metal-organic frameworks (MOFs)- and covalent organic frameworks (COFs)-based chiral luminophores (Supplementary Table 15). In sharp contrast, there is only negligible luminescence to be distinguished for the raw *d(l)*-thr ligands and molecular *d(l)*-Zn-thr complex (Supplementary Fig. 25), implying the unique collective chiroptical activity of assembled *d(l)*-Zn-HOIFs.

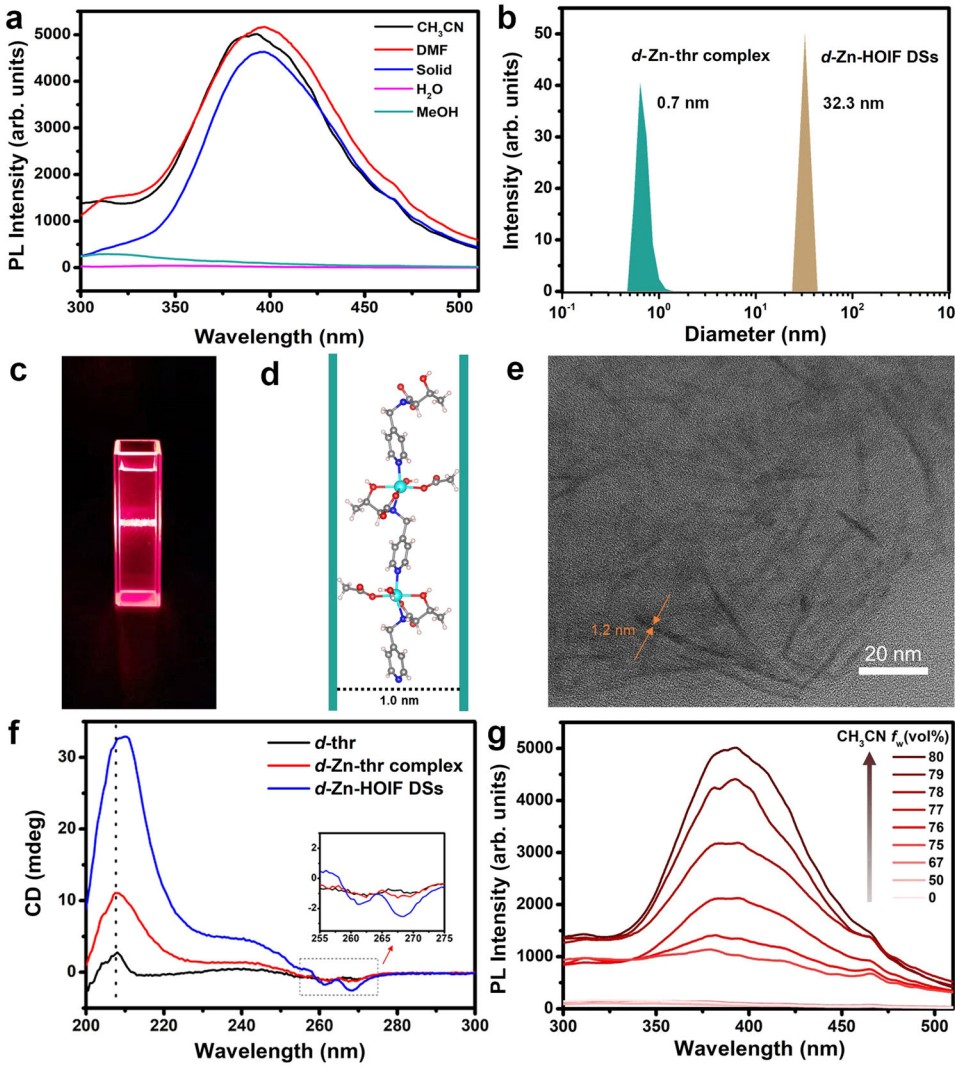

**Fig. 5 | Assembly induced chiroptical activities of *d*-Zn-HOIF. a** Fluorescence spectrum of *d*-Zn-HOIF NFs dispersed in different solvents at 265 nm excitation. **b** Dynamic light scattering (DLS) spectra of *d*-Zn-thr complex and *d*-Zn-HOIF DSs in H$_2$O. **c** Tyndall effect of *d*-Zn-HOIF DSs in H$_2$O. **d** Corresponding width of *d*-Zn-HOIF single chain (1.0 nm) in crystallography. **e** HR-TEM image of *d*-Zn-HOIF DSs in H$_2$O. **f** CD spectra of raw *d*-thr, molecular *d*-Zn-thr complex and *d*-Zn-HOIF DSs in H$_2$O. The inset magnifies the optical activity in the absorption window of aromatic pyridyl units. **g** Dynamically recovered fluorescence of *d*-Zn-HOIF NFs via aprotic solvent-induced reassembly.

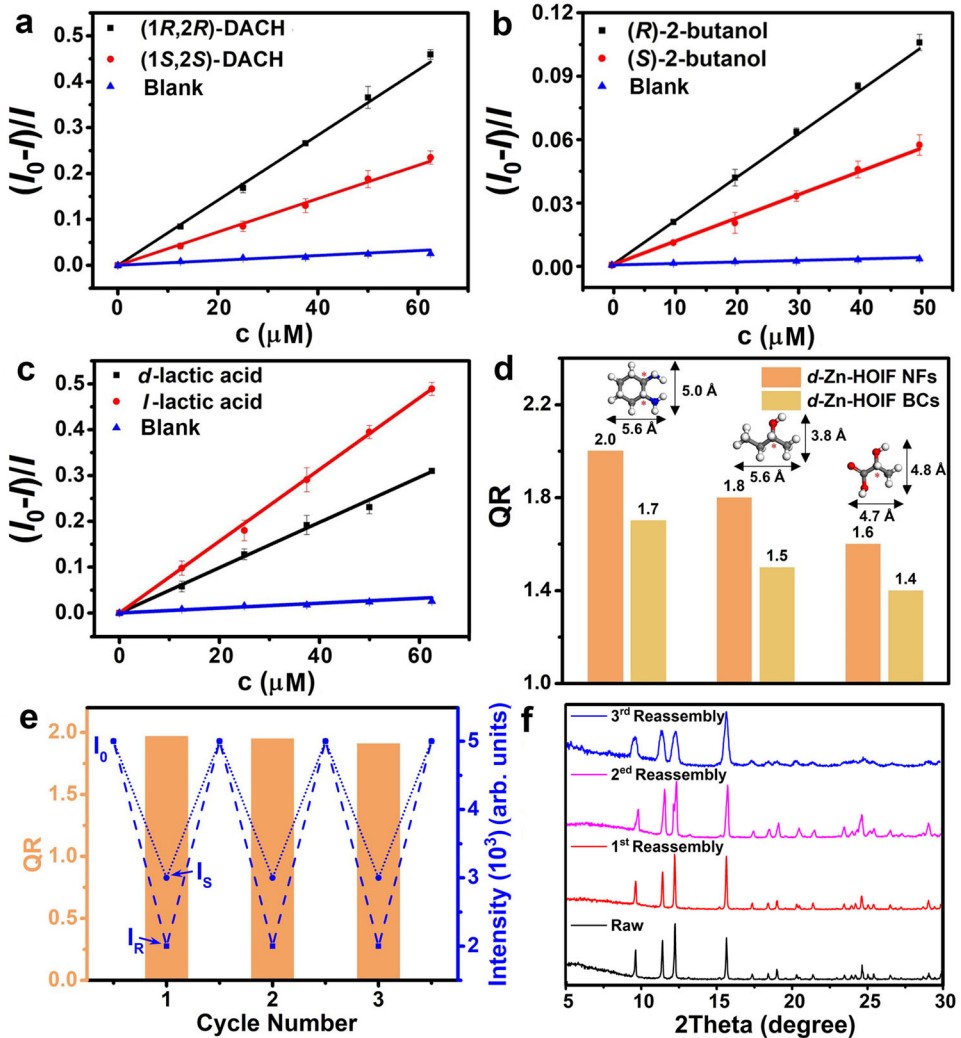

**Fig. 6 | Enantioselective recognition towards aliphatic substrates.** SV plots of *d*-Zn-HOIF NFs triggered by adding (**a**) (1*R*,2*R*)- and (1*S*,2*S*)-DACH, (**b**) (*R*)- and (*S*)−2-butanol, (**c**) *d*- and *l*-lactic acid. Blank curves present the tests free of analytes. Error bars represent standard deviations of three independent replicates.
**d** Enantioselective quenching ratio (QR) of tested chiral aliphatic compounds with various steric hindrances. The red asterisks represent the chiral centers. **e** Self-

purified and recyclable *d*-Zn-HOIF NFs for enantioselective recognition of (1*R*,2*R*)- and (1*S*,2*S*)-DACH during three consecutive cycles. $I_o$ stands for the initial FL intensity, while $I_R$ and $I_S$ represent the FL intensity after testing *R*- and *S*-enantiomer, respectively. **f** PXRD plots of reassembled *d*-Zn-HOIF NFs in comparison to the raw sample after each cycle.

## Dynamic assembly character and enantio-discriminability of chiral HOIFs

To explore the dynamic nature, the disassembled and reassembled *d*-Zn-HOIF NFs were systematically studied by fluorescence and CD spectroscopies. We suppose that *d*-Zn-HOIF NFs would display the dynamically quenched fluorescence via hydrogen bond donor-induced disassembly into *d*-Zn-HOIF discrete strands (DSs) accompanying with the reversibly recovered fluorescence via the aprotic solvent-induced reassembly (Fig. 4). Indeed, *d*-Zn-HOIF NFs possess a strong emission peak at 395 nm in aprotic acetonitrile (CH₃CN), dimethylformamide (DMF) or solid-state, whereas the heavily quenched fluorescence is observed in the strong protic solvent like water or methanol (MeOH) caused by competitive hydrogen bonding (Fig. 5a). Specifically, the *d*-Zn-HOIF DSs present random curly morphology (Supplementary Fig. 26) with an average length of 31.5 nm after dispersion in water, which correlates with the measured dynamic scattering size (DLS) of 32.3 nm in relative to the molecular size (0.7 nm) of *d*-Zn-thr complex (Fig. 5b). The strong Tyndall effect observed under illumination of red laser also indicates well dispersion nature of *d*-Zn-HOIF DSs in water (Fig. 5c). According to the resolved SXRD data, a

single helical chain features a width of 1.0 nm in crystallography (Fig. 5d), which is in good agreement with that of *d*-Zn-HOIF DSs (1.2 nm) estimated by HR-TEM (Fig. 5e, Supplementary Fig. 26). Similarly, Co-HOIFs and Ni-HOIFs are disassembled into corresponding single-unit chains after simple dispersion in water (Supplementary Figs. 27 and 28). In addition, the strong CD signal (Fig. 5f) at the same absorbance (Supplementary Fig. 29) for both *d(l)*-Zn-HOIF DSs claims their well-maintained helical configuration, which is distinct from the weak responses of raw *d(l)*-thr ligand and molecular *d(l)*-Zn-thr complex (inset in Fig. 5f and Supplementary Fig. 30). Altogether, the competitive hydrogen-bonding-induced disassembly into discrete helical strands instead of the coordination dissociation into metal-complexes occurs for *d*-Zn-HOIF NFs in the strong protic solvent. Excitingly, the quenched fluorescence of disassembled *d*-Zn-HOIF DSs can be recovered back to fluorescent *d*-Zn-HOIF NFs dynamically by adding the aprotic solvent (e.g. CH₃CN and acetone). One can see that the discernable fluorescence appears and then further increases its intensity upon the volume ratio of added CH₃CN over 75% (Fig. 5g). The PXRD pattern (Supplementary Fig. 31) of as-formed fiber-like precipitates (Supplementary

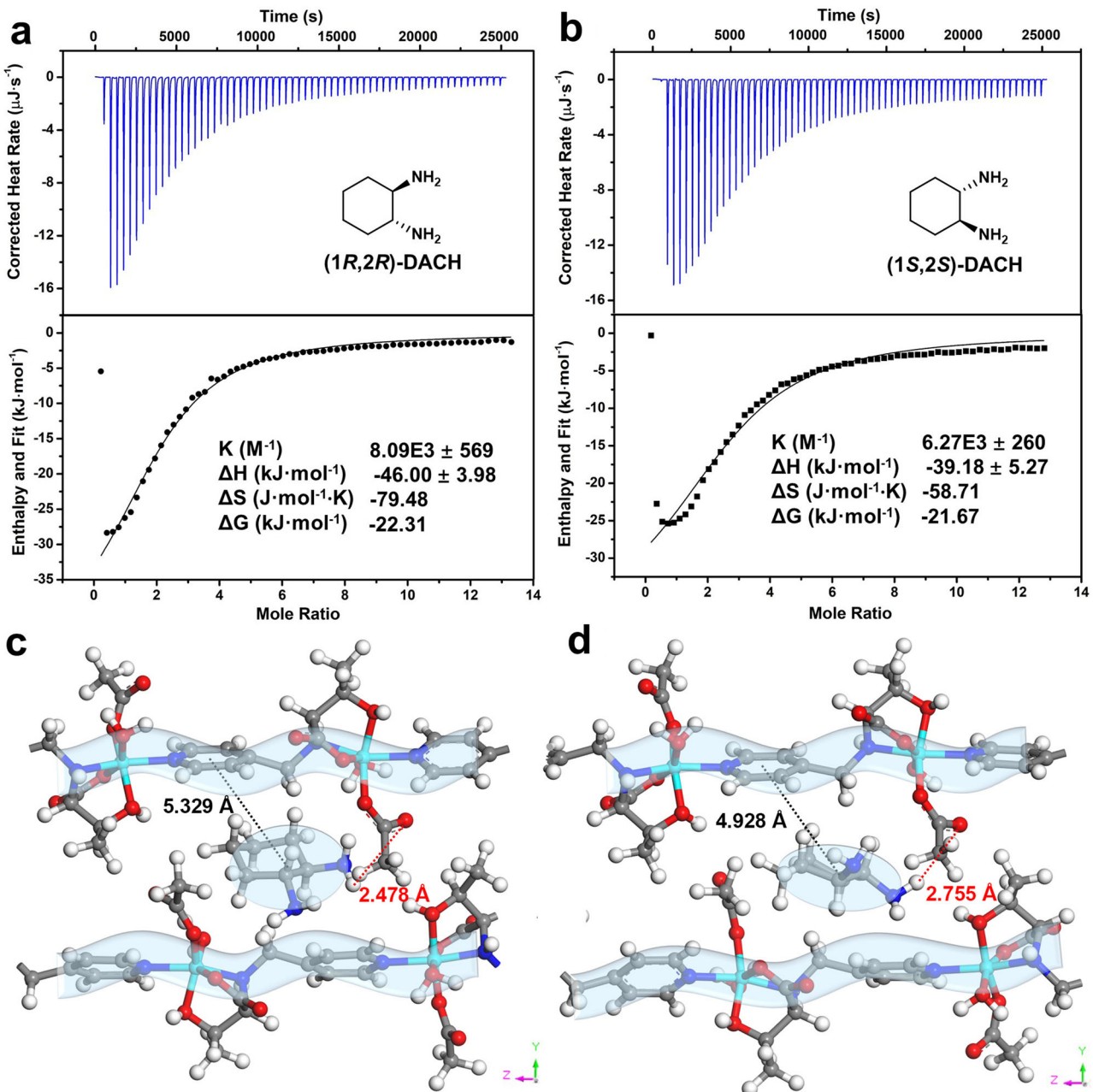

**Fig. 7 | Nano ITC analysis and theoretical calculation.** ITC curves of *d*-Zn-HOIF NFs with adding (**a**) (1*R*,2*R*)-DACH, (**b**) (1*S*,2*S*)-DACH. The thermochemical data are obtained by fitting corresponding titration curves by the independent model. Simulated binding interactions between *d*-Zn-HOIF NFs and (**c**) (1*R*,2*R*)-DACH,

(**d**) (1*S*,2*S*)-DACH with. The dashed lines indicate the hydrogen binding length between *d*-Zn-HOIF NFs and DACH. The black dashed lines indicate the steric hindrance between *d*-Zn-HOIF NFs and DACH.

Fig. 32) is consistent with that of simulated *d*-Zn-HOIF crystal, disclosing the regeneration of *d*-Zn-HOIF NFs via ordered reassembly rather than random aggregation[36].

The disassembly-reassembly dependent fluorescence property of *d*(*l*)-Zn-HOIF NFs enables them as a powerful platform for enantioselective recognition of chiral amine, alcohol, and carboxylic acid that may participate in the hydrogen bonding. Noteworthily, these substrates are biologically essential and synthetically valuable for catalysts, drugs, and pesticides[52,53]. Generally, *d*-Zn-HOIF NFs dispersed in ethanol (EtOH) are used as the probe considering good dispersibility and well-maintained fluorescence (Supplementary Fig. 33). The chiral analytes of opposite configurations are separately added for assessing the discrimination performance. The relationship between analyte

concentration and fluorescence intensity is analyzed by the well-known linear "Stern-Volmer" equation: $(I_0\text{-}I)/I = K_{sv} [c]$[54], where $I_0$ represents the initial fluorescence intensity free of adding chiral analytes, $I$ is the fluorescence intensity after adding chiral analytes, $[c]$ stands for the corresponding concentration of added analyte, and the constant $K_{sv}$ is the affinity between the analyte and *d*-Zn-HOIF NFs. Hence, the stereoselective recognition toward *R*- and *S*-analytes is quantitatively determined by the quenching ratio (QR), which is defined as the ratio of $K_{sv}(R)$ to $K_{sv}(S)$[55].

All the tested analytes quench the fluorescence of *d*-Zn-HOIF NFs but show distinct QR effects (Fig. 6, Supplementary Figs. 34–45 and Supplementary Table 16). We notice that though bearing the bulky steric hindrance, *d*-Zn-HOIF NFs only give a QR value of 1.2 and 1.1

towards the large-sized aromatic 1-phenylethanol (7.1 Å × 6.1 Å) and 1-phenylethylamine (7.1 Å × 6.1 Å), respectively, owing to their small chiral pore size (6.3 Å × 4.8 Å, Supplementary Fig. 46) inaccessible for the aromatic substrates. Then, we switch our efforts to chiral aliphatic substrates, which are more difficult to be discriminated due to their reduced steric hindrance in comparison to bulk aromatic substrates[56]. Significantly, $d$-Zn-HOIF NFs show the $K_{sv}$ value of $7.3 \times 10^3$ and $3.6 \times 10^3$ M$^{-1}$ toward (1$R$,2$R$)- and (1$S$,2$S$)-diaminocyclohexane (DACH) of appropriate molecular dimension of 6.3 Å × 4.8 Å, respectively, and thus the QR value considerably rises to 2.0 (Figs. 6a and d). More interestingly, $d$-Zn-HOIF NFs can even recognize chiral centers in the meso (1$R$,2$S$)-DACH, and the corresponding SV plot is well linearly fitted with a $K_{sv}$ of $5.5 \times 10^3$ M$^{-1}$, which is precisely between that of (1$R$,2$R$)- and (1$S$,2$S$)-DACH. Regarding aliphatic 2-butanol (5.6 Å × 3.8 Å) and lactic acid (4.7 Å × 4.8 Å) of even smaller molecular dimension, evident enantioselective discrimination is still acquired by $d$-Zn-HOIF NFs. The quenching slope of enantiomeric 2-butanol and lactic acid differs in Fig. 6b, c, and the corresponding QR is calculated to be 1.8 and 1.6, respectively. It deserves pointing out that $d$-Zn-HOIF NFs have higher QR performance than their bulk counterparts thanks to a higher exposure ratio of chiral sites and better dispersibility in solvents (orange bars versus yellow bars in Fig. 6d). Attempt to verify the real applications for substances of unknown chiral purities, DACH with different ee values were further tested for representation. The result shows that $d$-Zn-HOIF NFs exhibit a good linear relationship in the recognition of DACH with ee values ranging from −100% ($S$,$S$) to 0% (racemic) and further +100% ($R$,$R$) (Supplementary Fig. 47).

Beyond the high enantioselectivity, the dynamic nature of hydrogen bonding endows $d$-Zn-HOIF NFs with unique self-purifying and recycling capabilities. After testing, the disassembled $d$-Zn-HOIF NFs are reassembled by adding the aprotic solvent like CH$_3$CN. As manifested in Fig. 6e, the fluorescence of purified and regenerated $d$-Zn-HOIF NFs is completely recovered, and moreover these NFs are consecutively tested for at least three cycles without obvious QR decrease. The PXRD patterns (Fig. 6f) after each testing cycle validate the maintained crystal structures of the recovered $d$-Zn-HOIF NFs. Significantly, such efficient and recyclable enantioselective recognition via the dynamic disassembly-reassembly process is unavailable in conventional MOF-[12,57] and COF-based[51,55,58] platforms due to their much stronger linkages.

### Thermodynamics and theoretical calculation

The underlying mechanism of chirality bias afforded by $d$-Zn-HOIF NFs is decoded by Nanowatt isothermal titration calorimetry (ITC) together with theoretic simulation (Supplementary Figs. 48, 49 and Supplementary Table 17). Taking DACH for representative (Fig. 7a, b), the ITC-determined association constant ($K$) for (1$R$,2$R$)- isomer ($8.09 \times 10^3$ M$^{-1}$) is quite higher than $6.27 \times 10^3$ M$^{-1}$ for (1$S$,2$S$)-isomer. Clearly, the faster quenching rate acquired by (1$R$,2$R$)-isomer is contributed by its stronger hydrogen bonding with $d$-Zn-HOIF NFs. To be specific, the thermochemical data fitted from titration curves offer more negative enthalpy change (−46.00 kJ mol$^{-1}$) for (1$R$,2$R$)-isomer than −39.18 kJ mol$^{-1}$ for (1$S$,2$S$)-isomer, revealing the favorable enthalpic interaction between $d$-Zn-HOIF NFs and (1$R$,2$R$)-isomers. Meanwhile, the entropic reduction for both (1$R$,2$R$)-isomer (−79.48 J mol$^{-1}$ K) and (1$S$,2$S$)-isomer (−58.71 J mol$^{-1}$ K) stresses enthalpic but not entropic driving force and the more negative value for (1$R$,2$R$)-isomer presents the higher order degree stemming from its stronger association with $d$-Zn-HOIF NFs. Armed with experimental findings, molecular simulation was then carried out to testify the above findings. One can see that both (1$S$,2$S$)- and (1$R$,2$R$)-enantiomers are bound inside the micropores embedded by two neighboring homohelical chains (Fig. 7c, d). The careful inspection uncovers that (1$R$,2$R$)-isomer is more closely bound to the dangling acetate oxygen site via donating the chiral amine group (N-H···O-C, 2.478 Å). By contrast, (1$S$,2$S$)-isomer shows weaker N-H···O-

C interaction (2.755 Å) with the acetate oxygen caused by the larger steric hindrance (central-to-central distance of 4.928 Å) from the nearby left-twisted pyridyl ring compared with 5.329 Å for (1$R$,2$R$)-enantiomer (Fig. 7c, d). The calculated binding energy for (1$R$,2$R$)-DACH (−48.45 kJ mol$^{-1}$) is also more negative than −43.32 kJ mol$^{-1}$ for (1$S$,2$S$)-isomer, which supports the stronger binding interaction between (1$R$,2$R$)-DACH with $d$-Zn-HOIF NFs. Note that similarly preferred bonding interactions are found between ($R$)−2-butanol as well as $l$-lactic acid with $d$-Zn-HOIF NFs (Supplementary Table 17). In brief, the favorable hydrogen bonding between one isomer of the enantiomeric pairs with $d$-Zn-HOIF NFs is the origin of selective chiral recognition.

## Discussion

A serial of chiral HOIFs is elaborately designed and assembled via biomimicking the natural evolution process of proteins. The bioinspired HOIFs exhibit remarkable chiroptical activities together with dynamic hydrogen bond-mediated disassembly-reassembly features. Moreover, the HOIFs exhibit outstanding enantioselective recognition of many important aliphatic chiral substrates including amine, alcohol and carboxylic acid, which has been a long-standing challenge due to their imperceivable steric hindrance in comparison to conventionally studied aromatic substrates. The biomimetic strategy suggested here opens the avenue towards the design and construction of chiral hydrogen-bonded assembles with collective property and recoverable capability, which are highly keen for chiral separation, asymmetric catalysis, chiroptical device and other chirality-related applications.

## Methods

### Materials

Zinc acetate dihydrate (Zn(CH$_3$COO)$_2$·2H$_2$O, 99.99%) and 4-pydidinecarboxaldehyde (98%) were purchased from Shanghai Aladdin Bio-Chem Technology Co., Ltd. Cobalt acetate tetrahydrate (Co(CH$_3$COO)$_2$·4H$_2$O, 98%), nickel tetrahydrate acetate (Ni(CH$_3$COO)$_2$·4H$_2$O, 99.9%), sodium hydroxide (NaOH, 99%), $l$-threonine (99%), $d$-threonine (98%), dioctyl sulfosuccinate sodium salt (NaAOT, 95%), isooctane(99%), D$_2$O (99.9%), methanol (MeOH, analytical grade), ethanol (EtOH, analytical grade), acetonitrile (CH$_3$CN, analytical grade), (1$R$,2$R$)-, (1$R$,2$S$)- and (1$S$,2$S$)-diaminocyclohexane (DACH), ($R$)- and ($S$)-2-butanol and ($l$)- and ($d$)-lactic acid, ($R$)- and ($S$)−1-phenylethylamine, and ($R$)- and ($S$)-1-phenylethanol were bought from Beijing Innochem Science & Technology Co., Ltd. Sodium borohydride (NaBH$_4$, 97%) was obtained from Tianjin Kermel Chemical Reagent Co., Ltd. HCl (analytical grade) and acetone (99.5%) were supplied by Tianjin Fengchuan Chemical Reagent Co., Ltd. The deionized (DI) water used in our experiments was obtained from the laboratory water purification system. All chemicals were used directly without further purification.

### Synthesis of $N$-(4-pyridylmethyl)-$d$-threonine ligand ($d$-thr)

Typically, $d$-threonine (2.02 g, 17.0 mmol) and NaOH (0.68 g, 17.0 mmol) were dissolved in 10 mL deionized (DI) water. Then, 10 mL methanolic solution containing 4-pyridine formaldehyde (1.82 g, 17.0 mmol) was added within 10 min at 25 °C and under vigorous stirring. After the reaction at 25 °C for 12 h, the solution turned yellow and was transferred to an ice water bath. Immediately, NaBH$_4$ (0.8 g, 20 mmol) dissolved in 10 mL ice water was added by twice injection at a time interval of 2 h. The resultant solution was further stirred for 7 h and adjusted to a pH value of 5.5 by using 1.0 M HCl. After stirring at 25 °C for 12 h additionally, the solid mixture was collected by rotary evaporation of the solvent. The final product was extracted with 150 mL hot EtOH 3 times and collected by following EtOH evaporation. After being washed with 10 mL acetone, the white powder was obtained for further usage. Yield: 2.28 g, 64.4%.

$^{1}$H-NMR (D$_2$O, ppm): -CH$_3$ (1.25, d, 3H), -HN-CH (3.32, d, 1H), -CH (4.01, m, 1H), -CH$_2$ (4.30 dd, 2H), py-H (7.52, d, 2H), py-H (8.58, d, 2H)[13].C-NMR (solid sample, ppm): -CH$_3$ (19.59), -CH$_2$ (49.03), -CH-OH (66.19), -CH-HN (68.44), py-C (125.07), py-C (141.27), py-C (149.12), -COOH (171.38). FT-IR (KBr, cm$^{-1}$): $v_{OH}$, 3481; $v_{NH}$, 2966; $v_{as(CO2)}$, 1603; $v_{s(CO2)}$, 1421. Observed MS spectrum of protonated d-thr: m/z: 211.11, 100%; m/z: 212.11, 11.0%. Calculated: m/z: 211.11, 100%; m/z: 212.11, 10.8%. Specific rotation value: +25.8°. The NMR, MS, FT-IR and Specific rotation characterization results were available in Supplementary Figs. 1, 3, 5, 7 and Table 1, respectively.

### Synthesis of N-(4-pyridylmethyl)-l-threonine ligand (l-thr)

Similarly, l-threonine (2.02 g, 17.0 mmol) and NaOH (0.68 g, 17.0 mmol) were dissolved in 10 mL deionized (DI) water. Then, 10 mL methanolic solution containing 4-pyridine formaldehyde (1.82 g, 17.0 mmol) was added within 10 min at 25 °C and under vigorous stirring. After the reaction at 25 °C for 12 h, the solution turned yellow and was transferred to an ice water bath. Immediately, NaBH$_4$ (0.8 g, 20 mmol) dissolved in 10 mL ice water was added by twice injection at a time interval of 2 h. The resultant solution was further stirred for 7 h and adjusted to a pH value of 5.5 by using 1.0 M HCl. After stirring at 25 °C for 12 h additionally, the solid mixture was collected by rotary evaporation of the solvent. The final product was extracted with 150 mL hot EtOH 3 times and collected by following EtOH evaporation. After being washed with 10 mL acetone, the white powder was obtained for further usage. Yield: 2.28 g, 64.4%. Yield: 2.33 g, 65.9 %.

$^{1}$H-NMR (D$_2$O, ppm): -CH$_3$ (1.26, d, 3H), -HN-CH (3.33, d, 1H), -CH (4.02, m, 1H), -CH$_2$ (4.29 dd, 2H), py-H (7.53, d, 2H), py-H (8.59, d, 2H) $^{13}$.C-NMR (solid sample, ppm): -CH$_3$ (19.59), -CH$_2$ (49.03), -CH-OH (66.18), -CH-HN (68.43), py-C (125.06), py-C (141.25), py-C (149.11), -COOH (171.36). FT-IR (KBr, cm$^{-1}$): $v_{OH}$, 3480; $v_{NH}$, 2955; $v_{as(CO2)}$, 1601; $v_{s(CO2)}$, 1419. Observed MS spectrum of l-thr: m/z: 211.11, 100%; m/z: 212.11, 10.7%. Calculated: m/z: 211.11, 100%; m/z: 212.11, 10.8%. Specific rotation value: −24.1°. The NMR, MS, FT-IR and Specific rotation characterization results were available in Supplementary Figs. 2, 4, 6, 7 and Table 1, respectively.

### Synthesis of Zn(d-thr)(CH$_3$COO)H$_2$O bulk crystals (d-Zn-HOIF BCs)

In a typical synthesis, d-thr (84 mg, 0.4 mmol) was dissolved in 1.7 mL DI H$_2$O and Zn(CH$_3$COO)$_2$·2H$_2$O (180 mg, 0.8 mmol) was dissolved in 3.4 mL methanol separately. The two solutions were then mixed and sonicated for 2 min. The resultant transparent solution was adjusted to a pH value of 6.45 by using 1.0 M NaOH solution and was stood at room temperature for one week. Finally, colorless d-Zn-HOIF BCs suitable for single-crystal X-ray characterization were obtained by a simple filtration method. Yield: 88.6 mg, 62.4%. FT-IR (KBr, cm$^{-1}$): $v_{OH}$, 3563; $v_{NH}$, 3177; $v_{as(CO2)}$, 1639; $v_{s(CO2)}$, 1423. The FT-IR and SEM characterization results were available in Supplementary Figs. 9 and 14, respectively.

### Synthesis of Zn(l-thr)(CH$_3$COO)H$_2$O bulk crystals (l-Zn-HOIF BCs)

Similarly, l-thr (84 mg, 0.4 mmol) was dissolved in 1.7 mL DI H$_2$O and Zn(CH$_3$COO)$_2$·2H$_2$O (180 mg, 0.8 mmol) was dissolved in 3.4 mL methanol separately. The two solutions were then mixed and sonicated for 2 min. The resultant transparent solution was adjusted to a pH value of 6.45 by using 1.0 M NaOH solution and was stood at room temperature for one week. Finally, colorless d-Zn-HOIF BCs suitable for single-crystal X-ray characterization were obtained by a simple filtration method. Yield: 90.6 mg, 63.8%. FT-IR (KBr, cm$^{-1}$): $v_{OH}$, 3560; $v_{NH}$, 3175; $v_{as(CO2)}$, 1635; $v_{s(CO2)}$, 1420. The FT-IR and SEM characterization results were available in Supplementary Figs. 9 and 15, respectively.

### Synthesis of Co(d-thr)(CH$_3$COO)H$_2$O (d-Co-HOIF)

Typically, d-thr (84 mg, 0.4 mmol) was dissolved in 1.7 mL DI H$_2$O and Co(CH$_3$COO)$_2$·4H$_2$O (155 mg, 0.8 mmol) was dissolved in 3.4 mL methanol separately. The two solutions were then mixed and sonicated for 2 min. The resultant transparent solution was adjusted to a pH value of 5.95 by using 1.0 M NaOH solution and was stood at room temperature for 48 h. The PXRD characterization is shown in Supplementary Fig. 12.

### Synthesis of Co(l-thr)(CH$_3$COO)H$_2$O (l-Co-HOIF)

Similarly, l-thr (84 mg, 0.4 mmol) was dissolved in 1.7 mL DI H$_2$O and Co(CH$_3$COO)$_2$·4H$_2$O (155 mg, 0.8 mmol) was dissolved in 3.4 mL methanol separately. The two solutions were then mixed and sonicated for 2 min. The resultant transparent solution was adjusted to a pH value of 5.95 by using 1.0 M NaOH solution and was stood at room temperature for 48 h. The PXRD characterization is shown in Supplementary Fig. 12.

### Synthesis of Ni(d-thr)(CH$_3$COO)H$_2$O (d-Ni-HOIF)

Typically, d-thr (84 mg, 0.4 mmol) was dissolved in 1.7 mL DI H$_2$O and Ni(CH$_3$COO)$_2$·4H$_2$O (199 mg, 0.8 mmol) was dissolved in 3.4 mL methanol separately. The two solutions were then mixed and sonicated for 2 min. The resultant transparent solution was adjusted to a pH value of 6.25 by using 1.0 M NaOH solution and was stood at room temperature for 48 h. The PXRD characterization is shown in Supplementary Fig. 12.

### Synthesis of Ni(l-thr)(CH$_3$COO)H$_2$O (l-Ni-HOIF)

Similarly, l-thr (84 mg, 0.4 mmol) was dissolved in 1.7 mL DI H$_2$O and Ni(CH$_3$COO)$_2$·4H$_2$O (199 mg, 0.8 mmol) was dissolved in 3.4 mL methanol separately. The two solutions were then mixed and sonicated for 2 min. The resultant transparent solution was adjusted to a pH value of 6.25 by using 1.0 M NaOH solution and was stood at room temperature for 48 h. The PXRD characterization is shown in Supplementary Fig. 12.

### Synthesis of Zn(d-thr)(CH$_3$COO)H$_2$O nanofibers (d-Zn-HOIF NFs)

Typically, d-thr (84 mg, 0.4 mmol) and Zn(CH$_3$COO)$_2$·2H$_2$O (180 mg, 0.8 mmol) were dissolved into 3.6 mL DI H$_2$O and the resultant pH value was adjusted to 6.45 using 1.0 M NaOH solution. The above aqueous solution was homogeneously mixed with 20.0 mL 5.0 M NaAOT/isoctane oil phase via 5 min ultrasonication. The obtained microemulsion was subjected to a microwave reactor at 60 °C for 3 h. After cooling down to room temperature, white colloids were collected by 12850 x g centrifugation for 5 min and washed three times with EtOH. The obtained white precipitates were evacuated overnight at 25 °C for further usage. Yield: 72.3 mg, 50.9%. The SEM image for d-Zn-HOIF NFs is available in Supplementary Fig. 16.

### Synthesis of Zn(l-thr)(CH$_3$COO)H$_2$O nanofibers (l-Zn-HOIF NFs)

Similarly, l-thr (84 mg, 0.4 mmol) and Zn(CH$_3$COO)$_2$·2H$_2$O (180 mg, 0.8 mmol) were dissolved into 3.6 mL DI H$_2$O and the resultant pH value was adjusted to 6.45 using 1.0 M NaOH solution. Above aqueous solution was homogeneously mixed with 20.0 mL 5.0 M NaAOT/iso-octane oil phase via 5 min ultrasonication. The obtained microemulsion was subjected to a microwave reactor at 60 °C for 3 h. After cooling down to room temperature, white colloids were collected by 12850 x g centrifugation for 5 min and washed three times with EtOH. The obtained white precipitates were evacuated overnight at 25 °C for further usage. Yield: 73.6 mg, 51.8%. The SEM image for the product was available in Supplementary Fig. 17.

### Characterization

Single-crystal X-ray diffraction (SXRD) data were collected at 169.99(10) K with Cu Kα X-ray source (λ = 1.542 Å) on a SuperNova,

Dual, Cu at zero, AtlasS2 diffractometer. The analyses of SXRD data were carried out using the OLEX2 program. Powder X-ray diffraction (PXRD) patterns for *d(l)*-Zn-HOIFs were collected on a Bruker D8 DISCOVER diffractometer (Cu Kα, $\lambda = 1.542$ Å) installed with a panel detector at conditions of 40 kV and 30 mA. The other samples were recorded on a Bruker D8 ADVANCE diffractometer (Cu Kα, $\lambda = 1.542$ Å) operating at 40 kV and 40 mA. Fourier transform infrared (FT-IR) spectra were obtained on a Thermo Fisher Scientific Nicolet iS50 using the KBr disk method. Thermogravimetric analyses (TGA) were tested on a Netzsch TG 209 F3 Tarsus amid nitrogen flow with a temperature range of 40 °C to 840 °C and a heating rate of 10 °C min$^{-1}$. The fluorescence (FL) spectra were recorded on a Gangdong F-320 fluorescence spectrophotometer. Circular dichroism (CD) spectra were measured on the Applied Photophysice Chirascan V100 (Plus) spectrometer. Nuclear magnetic resonance (NMR) was performed on a Bruker AVANCE III HD 400 machine. Solid-state Nuclear magnetic resonance (NMR) was performed on a Bruker AVANCE NEO 600 M machine. Mass spectrometry (MS) was obtained on a Shimadzu LCMS-8040 machine. The circularly polarized dichroism (CPL) spectra were achieved on a JASCO CPL-300 spectrometer. The luminescence anisotropy factor, $g_{lum}$ factor, was calculated according to the equation $g_{lum} = [CPL/(32980/\ln 10)]/DC$, where *CPL* and *DC* represented the CPL intensity and PL intensity, respectively. High-resolution transmission electron microscopy (HR-TEM) and high-angle annular dark-field scanning transmission electron microscopy (HAAD-STEM) imaging were carried out on a JEM-F200 at a voltage of 200 KV. Scanning electron microscope (SEM) measurements were carried out on a Phenom ProX scanning electron microscope at 5.0 kV. The specific rotation was measured on a Shanghai Shenguang SGWzz−2 instrument at 589 nm (The D line of sodium). Dynamic light scattering (DLS) measurement was performed on a Malvern Zetasizer Pro. The DLS size for the *d*-Zn-thr complex was obtained by mixing Zn(CH$_3$COO)$_2$·2H$_2$O and *d*-thr in water immediately for testing.

## Enantioselective fluorescence recognition

The enantioselective fluorescence recognition towards five pairs of chiral enantiomers including (1*R*,2*R*)- and (1*S*,2*S*)-(DACH), (*R*)- and (*S*)-2-butanol, (*l*)- and (*d*)-lactic acid, (*R*)- and (*S*)−1-phenylethylamine, (*R*)- and (*S*)-1-phenylethanol was evaluated by as-synthesized HOIFs. Taking DACH enantiomers for representative, 1.25 mM (1*R*,2*R*)- or (1*S*,2*S*)-DACH ethanolic solution was sequentially added into 1.0 mL ethanolic solution containing 0.05 mg *d*-Zn-HOIF NFs or *d*-Zn-HOIF BCs. The final concentration of (1*R*,2*R*)- or (1*S*,2*S*)-DACH was 12.5, 25.0, 37.5, 50.0, and 62.5 μM, respectively. After stirring for 10 min in each addition cycle, the fluorescence evolution was immediately monitored by the FL spectrophotometer. The experiment was repeated three times in parallel and an averaged FL intensity was plotted as a function of the concentration of DACH enantiomer.

## Nano ITC titration

Nanowatt isothermal titration calorimetry (ITC) was performed on a TA NANO ITC System with stirring at 300 rpm and 25 °C. The titration data were analyzed with NanoAnalyze software (TA Instruments Inc.) and fitted using the independent model. Each injection of 4 μL 6 mM DACH enantiomer was pushed from a 250 μL microsyringe at an interval of 400 s into 0.14 mM *d*-Zn-HOIF NFs. Each injection of 4 μL 50 mM lactic acid enantiomer was pushed from a 250 μL microsyringe at an interval of 500 s into 2.8 mM *d*-Zn-HOIF NFs. Each injection of 4 μL 5 mM chiral enantiomer was pushed from a 250 μL microsyringe at an interval of 400 s into 0.14 mM *d*-Zn-HOIF NFs.

## Theoretical calculation

The pore dimension estimation for *d*-Zn-HOIF was performed through the atom volume and surfaces module integrated into the Material Studio software package[59], using the Connolly surface method and setting the probe radius of 1.0 Å. Molecular simulations were performed through the Dmol3 module integrated into the Material Studio software package. The calculations were performed using a periodic model of the skeleton using PBE, the generalized gradient approximation (GGA) exchange functional, double numerical plus D-functions (DND) polarization, and the solvation model of ethanol[60]. The geometry optimizations and related electronic structures of *d*-Zn-HOIF were carried out with the CP2K[61] package using PBE0[62]-ADMM[63] functional. The excited state energies and oscillator strengths were obtained from TD-DFT calculations with Tamm-Dancoff approximation[64]. The distribution of electrons and holes in the electron excitation process was calculated using Multiwfn (version 3.7)[65].

## Reporting summary

Further information on research design is available in the Nature Portfolio Reporting Summary linked to this article.

## Data availability

The authors declare that all data supporting the findings of this study are available within the paper and its supplementary information files or available from the corresponding author upon request. The crystallographic data generated in this study have been deposited in the Cambridge Crystallographic Data Centre with deposition numbers of CCDC 2209408 and 2209409. Source data are provided with this paper.

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

## Acknowledgements

The authors acknowledge financial support from the National Natural Science Foundation of China (22103055, J.G., 92056204, 21890381 and 21721002, Z.Y.T.), Science and Technology Plans of Tianjin (21ZYJDJC00050, J.G.), National Key Research and Development Program of China (2021YFA1200302, Z.Y.T.), Strategic Priority Research Program of Chinese Academy of Sciences (XDB36000000, Z.Y.T.) and Wenzhou Key Laboratory of Biomaterials and Engineering (WIU-CASSWCL21005, X.Q.G.). We would like to thank the Analytical & Testing Center of Tiangong University for the help of HR-TEM testing.

## Author contributions

J. G., M. Z., Z. T., and Y. L. proposed the research direction and guided the project. J. G. and Y. D. designed and performed most of the experiments, Y. D., and Y. Y. completed the isothermal titration calorimetry testing, Y. J., Z. Z., P. L., and X. G. helped to carry out the experimental tests. M. Z., F. L., Y. L., and H. C. participated in the data analyses and manuscript discussion. Y. D. and J. G. drafted the raw manuscript. J. G., M. Z., Z. T., and Y. L. revised the manuscript.

## Competing interests

The authors declare no competing interests.
