## [Peer Review File · Nature Communications]

REVIEWER COMMENTS

Reviewer #1 (Remarks to the Author):

In this article, Jun Guo et al. studied the chiroptical response of the chiral hydrogen-bond-induced organic-inorganic frameworks (HOIF). The submitted article includes an interesting concept of chirality induction from chiral amino acids to the metal frameworks via the biomimicking process. However, this paper lacks detailed analyses and it is questionable whether the synthesized HOIFs actually exhibit chiroptical response or not. Therefore, the reviewer thinks that it seems to be insufficient for publication in Nat. Comm. and significant improvement is necessary. The detailed comments are listed below.

1. The overall introduction lacks proper description of the advantage or need for the chiral HOIF.
2. In page 3, a detailed explanation for Figure S1 ~ S3 does not exist in neither main manuscript nor supporting information. The description on NMR and FT-IR analyses written in the supporting information is insufficient.
3. In page 4, the authors argued the applicability of the biomimetic assembly process to various types of chiral HOIF such as d(l)-Ni-HOIFs and d(l)-Co-HOIF (Figure S8 – S10). However, it seems inappropriate to suggest the applicability prior to XRD analysis of the d(l)-Zn-HOIF (Figure 2a), which is the main focus of this article. Furthermore, the authors should clearly explain the peak position of (001), (002), (012), (020), and (021) facets of the synthesized HOIFs because the interplanar distance of the crystal structures can be changed according to the metal components.
4. The authors conducted CD and CPL analyses to explore the chiroptical response of the synthesized chiral HOIFs. Notably, the CD signals are located at the UV region of 200 nm to 290 nm in Figure 2, but the HOIFs exhibit CPL response at visible range from 360 nm to 520 nm. This is quite a large wavelength difference between maximum CD signals (230 nm) and maximum CPL responses (430 nm). In principle, the CD and CPL emissions arise from the same energy states, however, the submitted spectra did not show the same energy states. The reviewer suspects that the CD spectra at the UV region show the chiroptical response of the chiral amino acid itself but the CPL spectra exhibit the chiroptical response of the chirality-induced metal frameworks. Therefore, the authors should provide the longer wavelength CD spectra of the synthesized HOIFs as well as the detailed understanding.
5. In Figure 5a, the peaks of PL spectra of the d-Zn-HOIF are located near 400 nm, but the maximum DC voltage in Figure 2e is located near 430 nm. Please explain the reason for the 30 nm red shift of the DC voltage spectra compared to the linear PL spectra.

Reviewer #2 (Remarks to the Author):

In this work, the authors reported the synthesis of a series of HOIF structures, which exhibit reversible enantioselective molecular recognition as well as chirality induced fluorescence. Specifically, the structures of d(l)-Zn-HOIFs are unambiguously elucidated by SC-XRD. Furthermore, to study their assembling-disassembling properties, d(l)-Zn-HOIF nano fibers were also synthesized. This is a very interesting work. Given the great importance and growing interest in organic and hybrid framework research, and the knowledge gained in this study that could help future development of novel chiral frameworks for potential molecular separation, sensing and catalysis applications, publication of this work on Nat. Commun. is recommended. The authors need to address the following issues:

1. In the experimental PXRD, why cannot the facet (011) be observed? If it's because of the low intensity, the authors should try to acquire higher quality data.
2. The ¹³C NMR, HRMS, and specific rotation value of d-thr and l-thr should be collected and reported.
3. The authors stated that the HOIFs they synthesized have enantioselective recognition. But the

possibility that the HOIFs could recognize chiral center cannot be ruled out. In other words, the meso compound: (1R, 2S)-DACH, should be tested.

4. In the main text, the authors mentioned they cultivate the crystals at 60 °C, but in the method part, they reported room temperature. Such inconsistency should be fixed.

5. In Scheme 1, the representation of "Zn(AC)2" is not accurate, they should either use "Zn(Ac)2" or "Zn(OAc)2" (recommended). And on the middle scheme of 1b, they clearly distinguished chelating bonds and covalent bonds, but on the right side, they didn't distinguish them.

6. In Figure 1, the authors should indicate which color represents which type of atom.

7. For the FT-IR representation, does the "CO2" indicates the carboxylic acid groups? If so, they should revise them to eliminate ambiguity.

8. The authors should check the consistency of the reference positions, if they are before punctuation or after punctuation.

Reviewer #3 (Remarks to the Author):

This manuscript reports that novel bioinspired HOIFs enable reversible disassembly into single helical strands and further reassembly into ordered frameworks, that are till untouched in both MOFs and COFs formed via strong bonding. Thanks to their assembly-induced chiroptical activities, more significantly, those chiral HOIFs have further exhibited high enantioselectivity and recoverable performances in chiral recognitions of aliphatic substances. Considering novelty and generality of the bioinspired strategy as well as the dynamic assembly-disassembly properties unusually seen in porous materials, this manuscript is recommended to be accepted by Nature Communications after a minor reversion.

(1) The universality of proposed biomimetic strategy is of great significance in constructing crystalline porous materials assembled from helical building blocks. Especially, the disassembled single helical chains in water have been directly observed via TEM characterizations. However, it is still suggested to further confirm whether similar single helical building blocks can be disassembled and observed for the isorecticular Co-HOIFs and Ni-HOIFs.

(2) In Figure 3f, strong and unique CD signals of d-Zn-HOIF DSs declare their helical configurations distinct from the d-thr ligand and d-Zn-thr complex. CD characterizations for corresponding l-enantiomers (including l-Zn-HOIF, l-thr and l-Zn-thr complex) are also suggested to be supplemented in order to solidify the result.

(3) Figure S19 shows that the reassembled d-Zn-HOIFs are stacked together and therefore their morphologies cannot be clearly discerned and compared with that of raw. Please give SEM images of better HOIF dispersions.

(4) Efficient chiral reorganization towards aliphatic substrates is meaningful. But in practice, chiral substrates are often enantiomeric mixtures with unknown ee values. Whether the chiral HOIFs are able to quantitatively discern the ee values of chiral aliphatic substrates.

(5) There are still few typo errors in the manuscript. For example, "elpised" is misspelled and should be "eclipsed" in the description of Figure 1d. It is recommended to further check the whole context.

Responses to Reviewer #1

General comment: In this article, Jun Guo et al. studied the chiroptical response of the chiral hydrogen-bond-induced organic-inorganic frameworks (HOIF). The submitted article includes an interesting concept of chirality induction from chiral amino acids to the metal frameworks via the biomimicking process. However, this paper lacks detailed analyses and it is questionable whether the synthesized HOIFs actually exhibit chiroptical response or not. Therefore, the reviewer thinks that it seems to be insufficient for publication in Nat. Comm. and significant improvement is necessary. The detailed comments are listed below.

Our response: We highly appreciate the referee for his/her positive recognition to the research concept proposed in our manuscript. Following the insightful comments offered by the referee, we have rigorously and thoroughly revised our manuscript, specially making a clear verification on the chiroptical activities of reported HOIFs. Please see our point-to-point responses and revisions listed below.

Comment 1: The overall introduction lacks proper description of the advantage or need for the chiral HOIF.

Our response: Thanks for the referee's constructive suggestion for highlighting the advantages of HOIFs reported in this manuscript. We have replenished additional comments and discussions in the introduction part.

Our revision: Please see the yellow highlighted discussions in the introduction part of the revised manuscript, which are pasted as follows for your convenience.

(Page 3)

In brief, chiral ligands and metal ions would combine into one-dimensional (1D) helical chains via enthalpy-favored coordination condensation. Sequentially, multiple complementary hydrogen bonds might facilitate the assembly of helical chains into two-dimensional (2D) grids or three-dimensional (3D) frameworks of well-defined chiral structures. **Distinct from conventional chiral metal-organic**

frameworks (MOFs) and covalent organic frameworks (COFs), the reported chiral HOIFs have exhibited unique assembled-induced chiroptical activities including the emerging circular polarized luminescence (CPL). Also, chiral HOIFs could be reversibly disassembled into single helical strands and be further reassembled into frameworks without losing ordered arrangement. Hence, they have been utilized as recoverable chiral platforms for enantioselective recognitions, which would be not implemented by chiral MOFs and COFs formed via strong linkages.

Comment 2: In page 3, a detailed explanation for Figure S1 ~ S3 does not exist in neither main manuscript nor supporting information. The description on NMR and FT-IR analyses written in the supporting information is insufficient.

Our response: Thanks very much for the referee's critical comments for improving the quality of our manuscript. In the original manuscript, the corresponding explanations for ¹H-NMR peaks were written in the Methods part on the main body of manuscript. Following the referee's kind suggestion, we have also added the detailed peak assignments in the revised supplementary information following each ¹H-NMR spectrum. Besides, we have further supplemented the ¹³C-NMR spectra of the corresponding chiral ligands accompanied with detailed explanations. With regard to the FT-IR spectra, we have not only offered the detailed assignments for the characteristic vibration peaks, but also supplemented the corresponding spectra of raw *d(l)*-threonine for a straightforward comparison.

Our revision: Please see the Supplementary Fig. 1-4 for the NMR spectra and the Supplementary Fig. 7 for the FT-IR spectra in the revised supplementary information. The corresponding explanations and discussions are available below each Figure. All the changes are pasted as follows for your convenience.

Supplementary Figure 1. $^1\text{H-NMR}$ spectrum of *d*-thr. (The peak centered at 4.79 ppm is the residual solvent peak of D_2O).

Peak assignments: $^1\text{H-NMR}$ (D_2O , ppm): $-\text{CH}_3$ (1.25, d, 3H), $-\text{HN-CH}$ (3.32, d, 1H), $-\text{CH}$ (4.01, m, 1H), $-\text{CH}_2$ (4.30 dd, 2H), py-H (7.52, d, 2H), py-H (8.58, d, 2H). The $^1\text{H-NMR}$ spectrum demonstrates the successful synthesis of *d*-thr in high purity.

Supplementary Figure 2. $^1\text{H-NMR}$ spectrum of *l*-thr. (The peak centered at 4.79 ppm is the residual solvent peak of D_2O).

Peak assignments: $^1\text{H-NMR}$ (D_2O , ppm): $-\text{CH}_3$ (1.26, d, 3H), $-\text{HN-CH}$ (3.33, d, 1H), $-\text{CH}$ (4.02, m, 1H), $-\text{CH}_2$ (4.29 dd, 2H), py-H (7.53, d, 2H), py-H (8.59, d, 2H). The $^1\text{H-NMR}$ spectrum demonstrates the successful synthesis of *l*-thr in high purity.

Supplementary Figure 3. $^{13}\text{C-NMR}$ spectrum of *d*-thr.

Peak assignments: $^{13}\text{C-NMR}$ (solid sample, ppm): $-\text{CH}_3$ (19.59), $-\text{CH}_2$ (49.03), $-\text{CH-OH}$ (66.19), $-\text{CH-HN}$ (68.44), py-C (125.07), py-C (141.27), py-C (149.12), $-\text{COOH}$ (171.38). The $^{13}\text{C-NMR}$ spectrum confirms the successful synthesis of *d*-thr in high purity.

Supplementary Figure 4. ^{13}C -NMR spectrum of *l*-thr.

Peak assignments: ^{13}C -NMR (solid sample, ppm): -CH₃ (19.59), -CH₂ (49.03), -CH-OH (66.18), -CH-HN (68.43), py-C (125.06), py-C (141.25, py-C (149.11), -COOH (171.36). The ^{13}C -NMR spectrum confirms the successful synthesis of *l*-thr in high purity.

Supplementary Figure 7. FT-IR spectra of *d*-thr (blue curve), *l*-thr (purple curve), *d*-threonine (red curve) and *l*-threonine (black curve).

Compared with *d(l)*-threonine precursor, the disappearance of H-N-H bending absorption peak centered at 2049 cm⁻¹ proves the successful formation of *d(l)*-thr ligands via Schiff-base condensation that is sequentially reduced to the secondary amine ($\nu_{\text{N-H}}$, 2966 cm⁻¹). Moreover, the asymmetric stretching vibration of carboxyl group of *d(l)*-thr ligands moves to 1603 cm⁻¹, which is also different from 1627 cm⁻¹ of raw *d(l)*-threonine.

Comment 3: In page 4, the authors argued the applicability of the biomimetic assembly

process to various types of chiral HOIF such as *d(l)*-Ni-HOIFs and *d(l)*-Co-HOIF (Figure S8 - S10). However, it seems inappropriate to suggest the applicability prior to XRD analysis of the *d(l)*-Zn-HOIF (Figure 2a), which is the main focus of this article. Furthermore, the authors should clearly explain the peak position of (011), (002), (012), (020), and (021) facets of the synthesized HOIFs because the interplanar distance of the crystal structures can be changed according to the metal components.

Our response: Thanks very much for the referee's constructive comments. First of all, we have adjusted the PXRD characterization part about *d(l)*-Zn-HOIFs just after their SXRD description. We then further demonstrate the applicability of proposed biomimetic assembly to other types of HOIFs such as isorecticular *d(l)*-Ni-HOIFs and *d(l)*-Co-HOIFs. We hope that such content adjustment can present a clear logic for our manuscript.

As for PXRD positions of the first five peaks namely (001), (002), (012), (020), and (021) crystal facets, we have summarized them in the below Table R1 and also indexed the interplanar distances specifically. One can see that the isorecticular *d(l)*-Ni-HOIFs present smaller facet distances such as $d_{(020)}$ and $d_{(002)}$ compared with the others, indicating their slightly shrunken crystallographic cells.

Table R1. The summarized five PXRD positions and corresponding facet distances.

Sample	2θ (°)	$d_{(011)}$ (Å)	2θ (°)	$d_{(002)}$ (Å)	2θ (°)	$d_{(012)}$ (Å)	2θ (°)	$d_{(020)}$ (Å)	2θ (°)	$d_{(021)}$ (Å)
d -Co-HOIF	7.85	11.25	9.60	9.20	11.41	7.75	12.47	7.09	13.35	6.62
l -Co-HOIF	7.88	11.21	9.58	9.22	11.44	7.73	12.50	7.07	13.36	6.62
d -Ni-HOIF	^a	-	9.74	9.07	11.64	7.59	12.70	6.96	13.60	6.50
l -Ni-HOIF	8.02	11.01	9.73	9.06	11.58	7.63	12.65	6.99	13.54	6.53
d -Zn-HOIF	7.81	11.31	9.62	9.18	11.43	7.72	12.34	7.16	13.25	6.67
l -Zn-HOIF	7.79	11.36	9.62	9.18	11.42	7.74	12.27	7.20	13.12	6.74

^aThe (011) facet is not obvious due to the preferred orientation of *d*-Ni-HOIF sample.

In addition, we have carefully analyzed the PXRD results of both *d(l)*-Ni-HOIFs and *d(l)*-Co-HOIFs by carrying out PXRD refinements (Fig. R1) in order to offer the corresponding cell parameters. As results collected in Table R2, one can clearly see the smaller cell parameters of *d(l)*-Ni-HOIFs especially along the *b*-axis and *c*-axis, due to the smaller ionic radius (0.69 Å) of Ni²⁺ compared with Co²⁺ (0.72 Å) and Zn²⁺ (0.74 Å).

Figure R1. The PXRD refinement for *d(l)*-Co-HOIFs and *d(l)*-Ni-HOIFs based on the Pawley method to index their cell parameters.

Table R2. The refined cell parameters based on Pawley refinement method.

Sample	Crystal system	Space group	a (Å)	b (Å)	c (Å)	α (°)	β (°)	λ (°)	r (Å)
d -Co-HOIF	Orthorhombic	P212121	6.14	14.17	18.40	90	90	90	0.72
l -Co-HOIF	Orthorhombic	P212121	6.14	14.17	18.40	90	90	90	0.72
d -Ni-HOIF	Orthorhombic	P212121	6.11	13.97	18.23	90	90	90	0.69
l -Ni-HOIF	Orthorhombic	P212121	6.13	14.02	18.32	90	90	90	0.69
d -Zn-HOIF	Orthorhombic	P212121	6.14	14.34	18.36	90	90	90	0.74
l -Zn-HOIF	Orthorhombic	P212121	6.14	14.41	18.37	90	90	90	0.74

Our revision: Please see the yellow-highlighted descriptions on page 4-5 in the revised

manuscript, which are pasted as follows for your convenience. The comprehensive PXRD characterizations for *d(l)*-Co-HOIFs and *d(l)*-Ni-HOIFs have also been replenished in the revised supplementary information (Supplementary Fig. 12-13 and Supplementary Table 14).

(Page 4-5)

adjacent sheets facilitate eclipsed AA stacking of those chiral sheets into the final 3D framework along the *a*-axis. The powder X-ray diffraction (PXRD) of *d(l)*-Zn-HOIF BCs exhibits the same diffraction pattern as the single crystal, with the first five peaks (i.e. 7.81°, 9.62°, 11.43°, 12.34° and 13.25°) assigned to the (011), (002), (012), (020) and (021) facet, respectively (Fig. 2a). Besides the crystallographic characterization, the nearly identical Fourier transform infrared (FT-IR) spectra (Supplementary Fig. 9) and thermogravimetric analysis (TGA) curves (Supplementary Fig. 10 and 11) of *d(l)*-Zn-HOIF BCs also support their same coordination environment and equivalent thermal stability. The proposed biomimetic assembly process is also applicable to varied types of chiral HOIFs, as demonstrated by successful constructions of isorecticular *d(l)*-Ni-HOIFs and *d(l)*-Co-HOIFs following the identical diffraction patterns to *d(l)*-Zn-HOIFs (Supplementary Fig. 12). In order to obtain the exact cell parameters, PXRD refinements (Supplementary Fig. 13) were further done with satisfactory agreement. As results summarized in Supplementary Table 14, those isorecticular HOIFs display nearly the same crystallographic parameters but with slightly shrunken cell dimensions for *d(l)*-Ni-HOIFs due to the smaller Ni²⁺ radius than others.

Comment 4: The authors conducted CD and CPL analyses to explore the chiroptical response of the synthesized chiral HOIFs. Notably, the CD signals are located at the UV region of 200 nm to 290 nm in Figure 2, but the HOIFs exhibit CPL response at visible range from 360 nm to 520 nm. This is quite a large wavelength difference between maximum CD signals (230 nm) and maximum CPL responses (430 nm). In

principle, the CD and CPL emissions arise from the same energy states, however, the submitted spectra did not show the same energy states. The reviewer suspects that the CD spectra at the UV region show the chiroptical response of the chiral amino acid itself but the CPL spectra exhibit the chiroptical response of the chirality-induced metal frameworks. Therefore, the authors should provide the longer wavelength CD spectra of the synthesized HOIFs as well as the detailed understanding.

Our response: Thanks to the referee's careful reading and valuable comment. First of all, we have collected the CD and absorption spectrum of *d(l)*-Zn-HOIFs from 300 to 500 nm (Fig. R2) by following his/her kind suggestion. There is no CD response in the longer wavelengths for *d(l)*-Zn-HOIFs as their absent absorbance in the corresponding region. In principle, CD signal arises from the absorbance difference between the left-handed and right-handed light (Circular dichroism: principles and applications. John Wiley & Sons, 2000, Page 29), therefore an observable CD signal originates from the premise of normal ground-state absorption. Unfortunately, the ligand-to-metal charge transfer (LMCT) appearing in visible wavelength region is forbidden for Zn^{2+} ($3d^{10}$) center without empty d-orbitals.

Figure R2. (a) CD and (b) normal absorption spectrum of *d(l)*-Zn-HOIFs in the wavelength region of 300 to 500 nm.

To clearly assign the observed CD response in the UV region, scrutiny on CD spectra of $d(l)$ -Zn-HOIFs, $d(l)$ -thr and $d(l)$ -Zn-thr complex has been done. As shown in Fig. R3, both $d(l)$ -Zn-HOIFs have displayed obviously distinct chiroptical responses in the absorption range of 200-300 nm compared with $d(l)$ -thr and $d(l)$ -Zn-thr complex. Specifically, the CD response within 240-280 nm merely observed from $d(l)$ -Zn-HOIFs (blue curves) are assigned to the pyridyl chromophore that is involved in the unique hierarchically helical structure of HOIFs. In sharp contrast, neglectable CD signal within this region reflects the insufficient chiroptical induction from only chiral carbon centers in both $d(l)$ -thr (black curves) and $d(l)$ -Zn-thr complex (red curves).

Figure R3. (a) CD spectra of d -thr, d -Zn-thr complex and d -Zn-HOIFs, (b) CD spectra of l -thr, l -Zn-thr complex and l -Zn-HOIFs, (c) and (d) corresponding absorbance spectra.

As mentioned above, the chiroptical response in the visible region is unavailable for $d(l)$ -Zn-HOIFs due to the forbidden LMCT. Fortunately, the isorecticular $d(l)$ -Co-HOIFs and $d(l)$ -Ni-HOIFs have indeed afforded the chiroptical response in the visible region, resulting from the not fully occupied d-orbitals of Co^{2+} ($3d^7$) and Ni^{2+} ($3d^8$) that enable the corresponding LMCT process. One can easily discern the strong and mirror-symmetric CD response in Fig. R4. Overall, we can therefore verify the observed chiroptical absorption stemming from chiral HOIFs rather than isolated chiral ligands

and complexes as well.

Figure R4. CD spectra of (a) *d(l)*-Co-HOIFs and (b) *d(l)*-Ni-HOIFs in the region of 300 to 800 nm, (c) and (d) corresponding absorbance spectra.

To understand the large Stokes shift between absorbance and luminescence, we have conducted the time-dependent density functional theory (TD-DFT) calculations at the PBE0-ADMM/DZVP-MOLOPT-SR-GTH level according to the single-crystal model. On the basis that absorption and emission share the same energy states, we take the calculated absorption spectra and related electronic transitions for demonstrations. As result shown in Fig. R5, the calculated absorption spectrum of *d*-Zn-HOIF presents the lowest absorption gap around 270 nm, which is in line with the experimentally obtained absorbance (blue curve in Fig. R3c) and declares the good reliability of our calculation in return.

Figure R5. The calculated absorption spectra of *d*-Zn-HOIF.

Furthermore, the electron transition density maps for the corresponding excited states are drawn in Fig. R6 for detailed analysis. Obviously, the involved four excited states coherently present an intramolecular charge transfer (ICT) transition between pyridyl and neighboring chemosphere. Principally, ICT would render a large red-shift for the emission band of a molecule caused by the concomitant dissipation of excitation energy, which can be as high as 1.0 eV (*ChemPhysChem* 2012, 13, 3714-3722 and *Sci. Adv.* 2022, 8, eabo3289).

Figure R6. The lowest four excited states of *d*-Zn-HOIF. The color shadow indicates the electron transition from the cyan to yellow region.

Besides the crucial role of ICT process, thermal relaxation of the excited state also

greatly contributes to the Stokes shift on basis of the Franck-Condon principle. Especially for a fluorophore substituted with unlocked or flexible groups via free single bonds (e.g., the C7-N8 single bond in our case, Fig. R7), large rotation and distortion relaxation around such single bond also result in large Stokes shift. As examples, the Stokes shift ($\Delta\lambda$) is over than 100 nm for the reported pyridine-derived dyes (*Angew. Chem. Int. Ed.* 2011, 50, 12214) and even as large as 157-210 nm for the 9-substituted pyronin analogues (*J. Org. Chem.* 2015, 80, 1299).

Figure R7. Schematic illustration of rotation and distortion relaxations of the excited state based on Franck-Condon principle.

Overall, the large Stokes shift ($\Delta\lambda = \text{ca. } 130 \text{ nm}$) observed in our case is reasoned to the prominent ICT-induced dissipation and thermal relaxation of the excited states. The reasonability of our reported data could be further confirmed by the reported results. For instance, the reported pyridyl-derived chiral MOF (*Adv. Mater.* 2020, 32, 2002914) showed an emission peak around 435 nm compared with the 270 nm of corresponding absorbance peak, giving a Stokes shift as large as 165 nm. Moreover, a reported Cd-MOF (*Polyhedron* 2021, 208, 115411) composed of similar pyridyl ligand showed an emission peak at 430 nm together with a large Stokes shift of ca. 160 nm.

Our revision: Please see the yellow-highlighted descriptions on page 6-7 in the revised manuscript, which are pasted as follows for your convenience. The corresponding

replenished Figures have been added in the revised supplementary information (Supplementary Fig. 18-21)

(Page 6-7)

Noteworthy, the CD response acquired by *d(l)*-Zn-HOIF NFs is obviously distinct from the isolated *d(l)*-thr ligands as well as *d(l)*-thr-Zn complex (Supplementary Fig. 18). Moreover, though chiroptical activity in the visible region is not observed for *d(l)*-Zn-HOIF NFs (Supplementary Fig. 19) due to the forbidden ligand-to-metal charge transfer (LMCT), isorecticular *d(l)*-Co-HOIF NFs and *d(l)*-Ni-HOIF NFs both display strong and mirror-symmetric CD response ranging from 200 to 800 nm (Supplementary Fig. 20 and 21). Inspired by the observed intriguing chiroptical absorptions, we further wonder whether the chiral luminescence stemming from pyridyl units emerges, which has been not reported yet among the hydrogen-bonded frameworks. Impressively, the circularly polarized luminescence (CPL) spectrum of *d*-Zn-HOIF NFs (Fig. 2d) exhibits considerably negative CPL signals in the window of 360 to 510 nm. And, similarly strong but positive CPL signals are discerned in the same wavelengths for *l*-Zn-HOIF NFs of the opposite chiral configuration. In relation to absorption constrained within UV region, the observed red-shifted CPL for *d(l)*-Zn-HOIF NFs is reasoned to the prominent charge transfer-induced dissipation⁴⁸⁻⁴⁹ and thermal relaxation of excited states⁵⁰⁻⁵¹ (Supplementary Fig. 22-24).

Comment 5: In Figure 5a, the peaks of PL spectra of the *d*-Zn-HOIF are located near 400 nm, but the maximum DC voltage in Figure 2e is located near 430 nm. Please explain the reason for the 30 nm red shift of the DC voltage spectra compared to the linear PL spectra.

Our response: Thanks very much for the referee's careful reading and kind reminding. After consulting the engineer, we have realized that the shift of the maximum wavelength from DC voltage spectra is likely caused by the light source attenuation of CPL instrument. Distinct from normal PL spectroscopy, CPL instrument is

commissioned to pick out the small difference (normally smaller than 0.1 %) between the left-handed circular polarization and the right-handed circular polarization of luminescence (*Nat. Commun.* 2023, 14, 1065). Hence, to obtain a satisfactory CPL spectrum needs a strong light source (i.e. a DC voltage around 0.5 V) generally. Once attenuation of the light source, the bandwidth of raster must be adjusted to a larger value for maintaining a high DC voltage but unfortunately at the compliance of decreased wavelength resolution, leading to possible discrepancy in PL peak and DC maximum. The inconsistency between the peak wavelength from CPL spectrum with that from normal PL spectrum are also seen in reported literatures. For example, the maximum emission peaks of PL and CPL were reported to be 438 nm and 420 nm for the chiral MOF, respectively (*Adv. Mater.* 2020, 32, 2002914). In another type of luminescent chiral MOF (*ACS Appl. Mater. Inter.* 2022, 14, 16435), the reported maximum emission peak positions were 387 nm and 410 nm from PL and CPL spectrum, respectively. In a recent literature about chiral COF (*J. Am. Chem. Soc.* 2022, 144, 7245), the maximum emission peaks of PL and CPL were 630 nm and 600 nm, respectively.

In order to fix this technical issue and afford more precise CPL spectra, we have retested the samples on the instrument with an updated light source. As expected, the recollected CPL spectra show the obviously decreased full width at half maximum (FWHM) of 40 nm compared with the 80 nm of the original spectra, an indication of the improved wavelength resolution. More importantly, the peak positions of both DC and CPL spectra are also obtained at 405 nm, basically in line with those (ca. 400 nm) from the normal PL spectra.

Our revision: Please see the updated Fig. 2d in the revised manuscript, which is pasted as follows for your convenience.

Figure 2 | Characterization of *d*(*l*)-HOIFs. a PXRD patterns of *d*(*l*)-Zn-HOIF NFs and BCs along with the simulated one. **b** Solid CD and **c** UV absorption spectra of *d*(*l*)-Zn-HOIF NFs. **d** CPL spectra of *d*(*l*)-Zn-HOIF NFs excited at 265 nm and **e** DC value standing for normal fluorescence intensity of *d*(*l*)-Zn HOIF NFs.

Responses to Reviewer #2

General comment: In this work, the authors reported the synthesis of a series of HOIF structures, which exhibit reversible enantioselective molecular recognition as well as chirality induced fluorescence. Specifically, the structures of *d(l)*-Zn-HOIFs are unambiguously elucidated by SC-XRD. Furthermore, to study their assembling-disassembling properties, *d(l)*-Zn-HOIF nano fibers were also synthesized. This is a very interesting work. Given the great importance and growing interest in organic and hybrid framework research, and the knowledge gained in this study that could help future development of novel chiral frameworks for potential molecular separation, sensing and catalysis applications, publication of this work on Nat. Commun. is recommended. The authors need to address the following issues:

Our response: We truly appreciate the referee for his/her valuable recommendation for our manuscript. In order to further improve the quality of our manuscript, we have rigorously followed all the insightful comments offered by the referee.

Comment 1: In the experimental PXRD, why cannot the facet (011) be observed? If it's because of the low intensity, the authors should try to acquire higher quality data.

Our response: We thank very much for the referee's constructive suggestion. The low peak intensity of facet (011) should be caused by the preferred orientation of *d(l)*-Zn-HOIFs with anisotropic 1D rod morphologies (*ACS Nano* 2019, 13, 7359). In order to eliminate the orientation and acquire better data, we have carefully grinded the samples and recollected diffraction data on a diffractometer installed with a plane detector.

Our revision: Please see the updated Fig. 2a in the revised manuscript, which presents obvious diffraction peaks for the (011) crystal facet. For the referee's convenience, it is also pasted herein.

Figure 2 | Characterization of *d(l)*-HOIFs. **a** PXRD patterns of *d(l)*-Zn-HOIF NFs and BCs along with the simulated one. **b** Solid CD and **c** UV absorption spectra of *d(l)*-Zn-HOIF NFs. **d** CPL spectra of *d(l)*-Zn-HOIF NFs excited at 265 nm and **e** DC value standing for normal fluorescence intensity of *d(l)*-Zn HOIF NFs.

2. The ^{13}C NMR, HRMS, and specific rotation value of *d*-thr and *l*-thr should be collected and reported.

Our response: We thank the referee for his/her valuable suggestions. We have replenished the ^{13}C -NMR, MS spectra and specific rotation values of *d(l)*-thr ligands in the revised supplementary information.

Our revision: Please see the Supplementary Fig. 3-4 (^{13}C -NMR spectra), the Supplementary Fig. 5-6 (MS spectra) and the Supplementary Table 1 containing specific rotation values in the revised supplementary information. Corresponding descriptions and discussions have also been added below each figure. The related Figures are also pasted as follows for your convenience.

Supplementary Figure 3. ^{13}C -NMR spectrum of *d*-thr.

Peak assignments: ^{13}C -NMR (solid sample, ppm): $-\text{CH}_3$ (19.59), $-\text{CH}_2$ (49.03), $-\text{CH}-\text{OH}$ (66.19), $-\text{CH}-\text{HN}$ (68.44), py-C (125.07), py-C (141.27), py-C (149.12), $-\text{COOH}$ (171.38). The ^{13}C -NMR spectrum confirms the successful synthesis of *d*-thr in high purity.

Supplementary Figure 4. ^{13}C -NMR spectrum of *l*-thr.

Peak assignments: ^{13}C -NMR (solid sample, ppm): $-\text{CH}_3$ (19.59), $-\text{CH}_2$ (49.03), $-\text{CH}-\text{OH}$ (66.18), $-\text{CH}-\text{HN}$ (68.43), py-C (125.06), py-C (141.25), py-C (149.11), $-\text{COOH}$

(171.36). The ^{13}C -NMR spectrum confirms the successful synthesis of *l*-thr in high purity.

Supplementary Figure 5. Mass spectrum of *d*-thr.

As shown above, the peak of $m/z = 211.11$ is the molecular ion peak of protonated *d*-thr and the peak of $m/z = 212.11$ is the corresponding isotopic signal. The consistency between the observed data with the calculated ones confirms the successful synthesis of *d*-thr ligand.

Supplementary Figure 6. Mass spectrum of *l*-thr.

Similarly, the peak of $m/z = 211.11$ is the molecular ion peak of protonated *l*-thr and the peak of $m/z = 212.11$ is the corresponding isotopic signal. The consistency between the observed data with the calculated ones confirms the successful synthesis of *l*-thr ligand.

Supplementary Table 1. The Specific rotation value of *d*-thr and *l*-thr ligand.

Sample	T (°C)	C (mg·ml ⁻¹)	L (mm)	λ (nm)	Specific rotation (°)
d -thr	30	1	100	589	+25.8
l -thr	30	1	100	589	-24.1

Where, T is the test temperature, C represents the sample concentration in water, L stands for the length of sample tube, and λ is the test wavelength of light. The *d*-thr and *l*-thr show specific rotation value of +25.8° and -24.1°, respectively.

Comment 3: The authors stated that the HOIFs they synthesized have enantioselective recognition. But the possibility that the HOIFs could recognize chiral center cannot be ruled out. In other words, the meso compound: (1*R*,2*S*)-DACH, should be tested.

Our response: We thank to the insightful comment from the referee. Following his/her kind advice, we have further tested the fluorescence quenching spectra and fitted corresponding SV plot of *d*-Zn-HOIF NFs towards the meso (1*R*,2*S*)-DACH. As shown in the supplementary Fig. 36, *d*-Zn-HOIF NF shows gradually decreased fluorescence upon increasing the concentration of (1*R*,2*S*)-DACH. The corresponding SV plot is well linearly fitted with a K_{sv} of $5.5 \times 10^3 \text{ M}^{-1}$ for (1*R*,2*S*)-DACH, which is just between that of (1*R*,2*R*)- and (1*S*,2*S*)-DACH. In a word, the reported *d*-Zn-HOIF NF demonstrates the recognition capacity towards chiral centers as well.

Our revision: Please see the Supplementary Fig. 36 in the revised supplementary information, which is pasted as follows for your convenience. The related discussion has also been added in the revised manuscript (yellow highlighted words on page 10).

Supplementary Figure 36. Fluorescence quenching spectra and corresponding SV plot of *d*-Zn-HOIF NFs by adding (1*R*,2*S*)-DACH. The concentration of (1*R*,2*S*)-DACH is changed from 0 to 62.5 μM.

According to the linearly fitted plot shown in Supplementary Fig. 36b, the K_{sv} of *d*-Zn-HOIF NFs towards meso (1*R*,2*S*)-DACH is $5.5 \times 10^3 \text{ M}^{-1}$, which is just between chiral (1*R*,2*R*)- and (1*S*,2*S*)-DACH. In addition, the LOD is calculated to be $3.2 \times 10^{-6} \text{ M}$.

Comment 4: In the main text, the authors mentioned they cultivate the crystals at 60 °C, but in the method part, they reported room temperature. Such inconsistency should be fixed.

Our response: Thanks very much for the referee's careful reading and kind reminding. We have revised the content in the main text.

Our revision: Please see the yellow highlighted description on page 4 of the main text in the revised manuscript, which is pasted as follows for your convenience.

(Page 4)

Then, $\text{Zn}(d(l)\text{-thr})(\text{CH}_3\text{COO})\text{H}_2\text{O}$ bulk crystals, denoted as $[d(l)\text{-Zn-HOIF BCs}]$, were cultivated in an aqueous solution containing $d(l)\text{-thr}$ and $\text{Zn}(\text{CH}_3\text{COO})_2 \cdot 2\text{H}_2\text{O}$ at room temperature for structure identification (part S2 in the supplementary information).

Comment 5: In Scheme 1, the representation of “ $\text{Zn}(\text{AC})_2$ ” is not accurate, they should either use “ $\text{Zn}(\text{Ac})_2$ ” or “ $\text{Zn}(\text{OAc})_2$ ” (recommended). And on the middle scheme of 1b, they clearly distinguished chelating bonds and covalent bonds, but on the right side, they didn't distinguish them.

Our response: We thank to the referee's valuable suggestion for improving the quality of our manuscript. We have revised the inaccurate representation of “ $\text{Zn}(\text{AC})_2$ ” to “ $\text{Zn}(\text{OAc})_2$ ” throughout the revised manuscript and the revised supplementary information. Moreover, chelating bonds and covalent bonds have been clearly distinguished in the whole Scheme 1b.

Our revision: Please see the yellow highlighted $\text{Zn}(\text{OAc})_2$ word in Scheme 1b of the revised manuscript as well as the revised supplementary information. The Scheme 1b has also been updated in the revised manuscript, which is pasted as follows for your convenience.

Scheme 1 | Biomimetic hierarchical assembly strategy proposed in this work. a Schematic

diagram of assembly evolution of natural proteins from chiral amino acids. The red dashed box represents the binding site involved in peptide condensation. **b** Structural diagram of chiral HOIFs proposed in this work via bioinspired assembly process. The blue arrow indicates the coordination of pyridyl units. The red dotted lines represent hydrogen bonding.

Comment 6: In Figure 1, the authors should indicate which color represents which type of atom.

Our response: Thanks very much for kind reminding. We have indicated the color of each type of element in the caption of Fig. 1. That is: C is gray; O is red; N is blue; H is light pink; Zn is cyan. Similarly, the color code for elements has also been added for *l*-Zn-HOIF presented in the revised supplementary information.

Our revision: Please see the added color codes in the caption of Fig. 1 and supplementary Fig. 8 in the revised manuscript and the revised supplementary information, respectively. All the changes are pasted as follows for your convenience.

Figure 1 | Assembly evolution of *d*-Zn-HOIF characterized by SXRD. a Asymmetric coordination mode of *d*-Zn-HOIF. The red asterisks represent chiral centers. **b** Assembled 1D helical chain with left-handed 2_1 screw and pitch of 18.4 Å along *c*-axis, **c** 2D grid and

d 3D framework via complementary hydrogen bonding. The red dotted lines represent hydrogen bonding. Element color: C is gray; O is red; N is blue; H is light pink; Zn is cyan.

Supplementary Figure 8. (a) Coordination mode of Zn(II) ion in *l*-Zn-HOIF. (b) 1D helical chain structure of *l*-Zn-HOIF projected along the *a*-axis. (c) 2D structure of *l*-Zn-HOIF projected within the *b*-*c* plane. (d) 3D framework of *l*-Zn-HOIF viewed down the *c*-axis. Element color: C is gray; O is red; N is blue; H is light pink; Zn is cyan.

Comment 7: For the FT-IR representation, does the “CO2” indicates the carboxylic acid groups? If so, they should revise them to eliminate ambiguity.

Our response: We thank to the referee again for his/her carefulness to improve the quality of our manuscript. The “CO2” in the original FT-IR spectra represented the carboxyl (C=O) vibration of carboxylic acid group in the origin manuscript but might be misunderstood in other meanings. Hence, we have replaced the “CO2” with “C=O” in the revised supplementary information. Other indications of FT-IR characteristic groups have also been revised to show clear presentation. A concise explanation has also been added below the FT-IR spectra.

Our revision: Please see the Supplementary Fig. 7 and 9 in the revised supplementary information, which are pasted as follows for your convenience.

Supplementary Figure 7. FT-IR spectra of *d*-thr (blue curve), *l*-thr (purple curve), *d*-threonine (red curve) and *l*-threonine (black curve).

Compared with *d(l)*-threonine precursor, the disappearance of H-N-H bending absorption peak centered at 2049 cm^{-1} proves the successful formation of *d(l)*-thr ligands via Schiff-base condensation that is sequentially reduced to the secondary amine ($\nu_{\text{N-H}}$, 2966 cm^{-1}). Moreover, the asymmetric stretching vibration of carboxyl group of *d(l)*-thr ligands moves to 1603 cm^{-1} , which is also different from 1627 cm^{-1} of raw *d(l)*-threonine.

Comment 8: The authors should check the consistency of the reference positions, if they are before punctuation or after punctuation.

Our response: We are grateful for the referee's careful reading and kind reminding. The positions of all citations have been double-checked and uniformly placed after the punctuation.

Our revision: Please see the yellow highlighted citation positions in the revised manuscript.

As a result, the collective performances of chiral assemblies are rarely reported along with chirality-related applications despite that there have been many

examples such as chiral gels,³¹⁻³³ polymers³⁴⁻³⁵ and aggregators³⁶⁻³⁸ of disordered structures.

Responses to Reviewer #3

General comment: This manuscript reports that novel bioinspired HOIFs enable reversible disassembly into single helical strands and further reassembly into ordered frameworks, that are till untouched in both MOFs and COFs formed via strong bonding. Thanks to their assembly-induced chiroptical activities, more significantly, those chiral HOIFs have further exhibited high enantioselectivity and recoverable performances in chiral recognitions of aliphatic substances. Considering novelty and generality of the bioinspired strategy as well as the dynamic assembly-disassembly properties unusually seen in porous materials, this manuscript is recommended to be accepted by Nature Communications after a minor revision.

Our response: We highly appreciate the referee for his/her positive recommendation for our manuscript. We have rigorously followed all the insightful comments offered by the referee in order to further improve the quality of our manuscript.

Comment 1: The universality of proposed biomimetic strategy is of great significance in constructing crystalline porous materials assembled from helical building blocks. Especially, the disassembled single helical chains in water have been directly observed via TEM characterizations. However, it is still suggested to further confirm whether similar single helical building blocks can be disassembled and observed for the isorecticular Co-HOIFs and Ni-HOIFs.

Our response: We are grateful for this valuable suggestion. Following the referee's kind comment, we have replenished the corresponding TEM characterizations for both Co-HOIFs and Ni-HOIFs. As shown in below Fig. R8, *d*-Co-HOIF DSs present similar curl-like morphologies with a width of 1.2 nm and a length of 15.5 nm. Moreover, *d*-Ni-HOIF DSs (Fig. R9) also exhibit curl-like morphologies with a width of 1.3 nm and a length of 19.9 nm. It is worth pointed out that the measured widths of both *d*-Co-HOIF DSs and *d*-Ni-HOIF DSs are in good agreement with the width of single-strand chain in crystallography. Consequently, both isorecticular Co-HOIF DSs and Ni-HOIF

DSs are composed of single helical stands similar to Zn-HOIF DSs.

Figure R8. (a) HR-TEM image of *d*-Co-HOIF DSs in H₂O. (b) HAAD-STEM image of *d*-Co-HOIF DSs in H₂O. (c) Diameter distribution of *d*-Co-HOIF DSs. (d) Length distribution of *d*-Co-HOIF DSs.

Figure R9. (a) HR-TEM image of *d*-Ni-HOIF DSs in H₂O. (b) HAAD-STEM image of *d*-Ni-HOIF DSs in H₂O. (c) Diameter distribution of *d*-Ni-HOIF DSs. (d) Length distribution of *d*-Ni-HOIF DSs.

Our revision: According to the referee's kind suggestion, the TEM characterizations have been added as the supplementary Fig. 27 and 28, and the corresponding discussions have also been added on page 8 in the revised manuscript (please see the yellow highlighted words).

Comment 2: In Figure 3f, strong and unique CD signals of *d*-Zn-HOIF DSs declare their helical configurations distinct from the *d*-thr ligand and *d*-Zn-thr complex. CD characterizations for corresponding l-enantiomers (including *l*-Zn-HOIF, *l*-thr and *l*-Zn-thr complex) are also suggested to be supplemented in order to solidify the result.

Our response: We greatly appreciate the constructive comment from the referee. The CD responses of raw *l*-thr, molecular *l*-Zn-thr complex and *l*-Zn-HOIF DSs have been added in the revised supplementary information (Supplementary Fig. 30). Indeed, they have exhibited the opposite CD signals with respect to the corresponding *d*-enantiomers shown in Fig. 3f. Compared with *l*-thr ligand and *l*-Zn-thr complex, the much stronger CD signals acquired by *l*-Zn-HOIF DSs combined with the unique CD response at wavelengths of 255-275 nm stemming from the pyridyl unit declare their hierarchically helical structures. In conjugation with the results afforded by *d*-enantiomers (Fig. 3f), the disassembled Zn-HOIF DSs are unambiguously verified to well maintain the helical configurations.

Our revision: According to the referee's kind suggestion, the CD spectra of raw *l*-thr, molecular *l*-Zn-thr complex and *l*-Zn-HOIF DSs have been added in the revised supplementary information (Supplementary Fig. 30), which are pasted as follows for your convenience. Additional analysis and discussion have also been inserted on page 8 of the revised manuscript (please see the yellow highlighted words).

Supplementary Figure 30. (a) CD spectra and (b) UV absorption of raw *l*-thr, molecular *l*-Zn-thr complex and *l*-Zn-HOIF DSs in H₂O under the same concentration (5 mM) of *d*-thr motif. The inset magnifies the optical activity in the absorption window of aromatic pyridyl units.

Compared with CD responses of *l*-thr and *l*-Zn-thr complex, the much stronger CD signals together with the characteristic CD signals (255-275 nm) of pyridyl units verify the well-maintained 1D helical structures of *l*-Zn-HOIF DSs in H₂O.

Comment 3: Figure S19 shows that the reassembled *d*-Zn-HOIFs are stacked together and therefore their morphologies cannot be clearly discerned and compared with that of raw. Please give SEM images of better HOIF dispersions.

Our response: We thank the referee's insightful suggestion. The SEM image of reassembled *d*-Zn-HOIFs has been recaptured and used to replace the original one. The 1D morphology with better dispersion of reassembled *d*-Zn-HOIFs can be clearly seen from the supplementary Fig. 32 in the revised supplementary information.

Our revision: According to the referee's kind suggestion, the SEM image of the reassembled *d*-Zn-HOIFs has been updated for a clear comparison to the raw sample. Please see Supplementary Fig. 32 in the revised supplementary information, and the related discussions have also been added on page 35 of the revised supplementary information, which are pasted as follows for your convenience.

Supplementary Figure 32. SEM image of reassembled *d*-Zn-HOIF precipitates.

As shown in Supplementary Fig. 32, the reassembled *d*-Zn-HOIF precipitates show the 1D rod morphology clearly.

Comment 4: Efficient chiral reorganization towards aliphatic substrates is meaningful. But in practice, chiral substrates are often enantiomeric mixtures with unknown ee values. Whether the chiral HOIFs are able to quantitatively discern the ee values of chiral aliphatic substrates.

Our response: We thank the referee's valuable advice for further expanding the practical applications of reported chiral HOIFs. Taking 1,2-DACH as the representative example, the fluorescence quenching spectra of *d*-Zn-HOIF NFs upon adding 1,2-DACH with different ee values (namely +100 %, +50 %, 0 %, -50 % and -100 %) have been monitored. Remarkably, the quenched fluorescent intensity (I_0-I/I) presents a good linear relationship to the corresponding ee value according to the fitted

SV plot. Consequently, the reported *d*-Zn-HOIF NFs are competent to quantitatively identify the ee values of chiral 1,2-DACH.

Our revision: According to the referee's kind suggestion, the quantitative relationship between corresponding fitted K_{sv} with ee value of 1,2-DACH afforded by *d*-Zn-HOIF NFs has been added in the revised supplementary information (Supplementary Fig. 47), which is pasted as follows for your convenience. Additional analysis and discussion have also been added on page 10 of the revised manuscript (please see the yellow highlighted words).

Comment 5: There are still few typo errors in the manuscript. For example, "elpised" is misspelled and should be "eclipsed" in the description of Figure 1d. It is recommended to further check the whole context.

Our response: We thank the referee's careful reading and kind reminding. The misspelling of "elpised" has been corrected to "eclipsed" in the revised manuscript. In addition, the entire manuscript and supplementary information have been carefully

examined and revised if applicable.

Our revision: Please see that the yellow highlighted word "eclipsed" on page 4 of the revised manuscript.

REVIEWERS' COMMENTS

Reviewer #1 (Remarks to the Author):

The authors have well addressed all the concerns raised by reviewer. The manuscript is now suitable for publication without further revision.

Reviewer #2 (Remarks to the Author):

The authors have carefully addressed the reviewers' questions and concerns. The quality of the manuscript has been further improved. Now publication of this work on Nat. Commun. is recommended.

Reviewer #3 (Remarks to the Author):

The authors have addressed the issues that I raised, and the quality of the manuscript has been improved. It could be accepted as it is.